# Potassium dependent rescue of a myopathy with core-like structures in mouse

M Gartz Hanson[1], Jonathan J Wilde[1,2], Rosa L Moreno[3], Angela D Minic[1], Lee Niswander[1,2]*

[1]Department of Pediatrics, University of Colorado, Anschutz Medical Campus, Aurora, United States; [2]Graduate Program in Cell Biology, Stem Cells and Development, University of Colorado, Anschutz Medical Campus, Aurora, United States; [3]Department of Physiology, University of Colorado, Anschutz Medical Campus, Aurora, United States

**Abstract** Myopathies decrease muscle functionality. Mutations in ryanodine receptor 1 (RyR1) are often associated with myopathies with microscopic core-like structures in the muscle fiber. In this study, we identify a mouse RyR1 model in which heterozygous animals display clinical and pathological hallmarks of myopathy with core-like structures. The RyR1 mutation decreases sensitivity to activated calcium release and myoplasmic calcium levels, subsequently affecting mitochondrial calcium and ATP production. Mutant muscle shows a persistent potassium leak and disrupted expression of regulators of potassium homeostasis. Inhibition of $K_{ATP}$ channels or increasing interstitial potassium by diet or FDA-approved drugs can reverse the muscle weakness, fatigue-like physiology and pathology. We identify regulators of potassium homeostasis as biomarkers of disease that may reveal therapeutic targets in human patients with myopathy of central core disease (CCD). Altogether, our results suggest that amelioration of potassium leaks through potassium homeostasis mechanisms may minimize muscle damage of myopathies due to certain RyR1 mutations.

*For correspondence: Lee.
Niswander@ucdenver.edu

**Competing interests:** The authors declare that no competing interests exist.

**Reviewing editor**: Giulio Cossu, University of Manchester, United Kingdom

## Introduction

Myopathies due to mutations in RyR1 are inherited and currently incurable. Congenital myopathies are characterized by hypotonia and delay of motor development with weakness in skeletal muscles. In normal muscle, nuclei are located at the periphery of the myofibers and the mitochondria reside throughout the myofiber. Muscle biopsies of patients with myopathies and mutations in ryanodine receptor type 1 (RyR1) show centralized nuclei as well as disorganized areas in the center of the myofiber, called cores, that lack mitochondria and are devoid of metabolic activity, reflective of cellular damage. The most common RyR1-associated myopathy is central core disease (CCD) (**Wu et al., 2006**; **Jungbluth, 2007**). In addition, RyR1 mutations in muscle cause diseases such as multi-mini core disease (MMD), hypokalemic periodic paralysis, and malignant hyperthermia (MH), although the phenotype/genotype relationship is unclear (**Fujii et al., 1991**; **Marchant et al., 2004**; **Treves et al., 2008**; **Wilmshurst et al., 2010**).

Muscle movement is produced through excitation–contraction (E–C) coupling elicited by t-tubule conductance of the muscle action potential. Depolarization stimulates the dihydropyridine receptor (DHPR, $Ca_V1.1$) to mechanically open RyR1 channels in the sarcoplasmic reticulum (SR), which release intracellular calcium stores to flood the myofiber with calcium and initiate muscle contraction (**Numa et al., 1990**). During contractions, muscle cells influx $Na^+$ and efflux $K^+$, leading to depolarization of

**eLife digest** Skeletal muscle covers our skeleton and allows us to move around. One disorder that leads to weakness in skeletal muscle—known as central core disease—can leave affected infants 'floppy' and delay the development of motor skills such as sitting, crawling, and walking. While no cure or treatment currently exists for the disease, researchers have found that most cases are connected to a mutation in the gene that makes a protein called ryanodine receptor type 1 (RyR1).

RyR1 belongs to a family of proteins that create channels for the controlled release of calcium ions from stores within cells. For muscle cells to contract, calcium ions must be released from these internal stores at the same time as potassium ions leave the cells. To relax the muscle cells, calcium ions are pumped back into the internal stores and potassium ions are taken back into the cell. Previous studies have established a role for RyR1 in the contraction of skeletal muscle, but the precise molecular details are not known.

Here, Hanson et al. studied mice that had symptoms of central core disease due to a mutation in the gene that makes RyR1. The muscle weakness in these mice was caused by defects that hindered the release of calcium ions from internal stores and leakage of potassium ions from the muscle cells.

The experiments reveal that a high-potassium diet alleviates the symptoms of disease in the mice by increasing the amount of potassium surrounding the muscle cells. Treatment with an existing drug called glibenclamide also reversed the disease symptoms by reducing the leakage of potassium ions from the cells.

Hanson et al. also found several genes involved in controlling potassium ion levels in cells that could act as indicators of the presence of the disease. These findings suggest that therapies targeting the control of potassium ion levels in muscle cells could minimize muscle damage in patients with central core disease.

the membrane, a rise in extracellular $K^+$, and loss of excitability. The myoplasmic loss of potassium is stabilized via increased potassium conductance from the opening of voltage-gated potassium channels (*Lindinger et al., 2001*; *Nielsen et al., 2004*) and the activation of sodium–potassium ATPase ($Na^+$, $K^+$-ATPase) transport (*Clausen, 2013b*).

During muscle contractions, the enzymes $Na^+/K^+$-ATPase, $Ca^{2+}$-ATPase, and myosin ATPase consume most of the stored ATP. Myosin ATPase is considered to account for the majority of ATP consumption at 60–70% (*Smith et al., 2005*). $Ca^{2+}$-ATPase is assumed to account for the majority of remaining ATP, while $Na^+$–$K^+$-ATPase can consume up to 10% of available ATP (*Homsher, 1987*; *Clausen et al., 1991*; *Barclay et al., 2007*). As ATP levels decrease during repeated muscle contractions and fatigue, ATP-sensitive $K_{ATP}6.2$ channels open causing potassium efflux, membrane depolarization, and inhibition of the action potential (*Gong et al., 2003*).

Thus, potassium homeostasis mechanisms preserve E–C coupled release of internal calcium stores, protect against muscle fatigue, and prevent excessive $K^+$ loss (*Clausen, 1986*; *Cifelli et al., 2008*) and may protect the muscle during ATP loss (*Gong et al., 2003*).

In this study, we explore the function of RyR1 as it pertains to mechanisms of muscle weakness and potassium homeostasis. Through a forward genetic screen, we identified a mutation in *Ryr1* that causes muscle weakness and myopathic pathological hallmarks in heterozygous mice. Furthermore, we examined the potential of this animal model for clinical and therapeutic studies. Mutant *Ryr1* muscle showed decreased concentrations of $Ca^{2+}$ and $K^+$ in the myoplasm and an increased permeability for $K^+$ in muscles at rest. In vitro, the potassium leak can be rectified through the application of higher concentrations of $K^+$ or inhibition of $K_{ATP}$ channels. In vivo, the muscle weakness and myopathic pathology can be reversed through increased interstitial potassium and pharmacological inhibition of $K_{ATP}$ channels either by diet or FDA-approved drugs, suggesting defects in $K^+$ homeostasis mechanisms as possible myopathological hallmarks due to mutations in RyR1. $K_{ATP}6.2$ channels and other regulators of potassium transport are misexpressed in mutant muscle and may partly underlie the disease phenotype in this myopathic animal model. Finally, we find that human patients with CCD show dysregulation of genes involved in the control of potassium homeostasis and suggest that these may serve as important disease biomarkers.

## Results

### Identification of RyR1 mutation that induces a myopathy with core-like structures in mice

Through a forward genetic screen in mice, we identified a mutation in which heterozygous mutant mice display muscle weakness. Sequencing identified an E4242G change in exon 93 of *Ryr1* (*Figure 1A*) outside of the channel region of RyR1. This allele is named *RyR1^m1Nisw*, but here we will refer to it as *Ryr1^AG*. Gene identification was confirmed by a lack of complementation in a genetic cross between *RyR1^AG* and *RyR1^tm1TAlle* allele (see 'Materials and methods'). RyR1 mutations in humans are associated with several congenital myopathies including CCD, multi-mini core disease, hypokalemic periodic paralysis, and malignant hyperthermia (MH) (*Fujii et al., 1991*; *Marchant et al., 2004*; *Jungbluth, 2007*; *Treves et al., 2008*; *Wilmshurst et al., 2010*). 1-month, 2-month, and 1-year old *Ryr1^AG/+* mice showed a significant decrease in grip strength and considerable deficits on a wire hanging task compared to *Ryr1^+/+* mice (*Figure 1B,C*). Malignant hyperthermia test was performed by placing *Ryr1^+/+* and *Ryr1^AG/+* mice in a 41°C humidified incubator for 30 min. There was no sign of muscle rigidity or spasticity or seizure activity for any of the animals. To test for sensitivity to halogenated anesthetics, anesthetic isoflurane was administered to *Ryr1^AG/+* mice at ~$5.5 \times 10^{-5}$ ml/cm$^3$ for a maximum of 30 min exposure. *Ryr1^AG/+* mice showed no MH response after exposure to isoflurane. The rectal temperature in *Ryr1^AG/+* mice after isoflurane exposure was similar to wild-type littermates [*Ryr1^AG/+* was 35.6 ± 0.9°C at 2 months (n = 5) and 35.9 ± 0.6°C at 6 months (n = 5) compared with *Ryr1^+/+* 35.4 ± 0.8°C and 35.8 ± 0.7°C at 2 and 6 months respectively (n = 5 for both)]. Thus, heterozygous mutant mice show muscle weakness, the clinical definition of a myopathy.

In typical muscle, nuclei are located at the periphery of the myofiber and mitochondria are present throughout. Muscle biopsies of CCD patients show internalized, centrally located nuclei, disorganized cores that lack mitochondria, and reduced metabolic activity (visualized with cytochrome oxidase (COX) and nicotinamide adenine dinucleotide hydride-tetrazolium reductase (NADH-TR) staining) (*Wu et al., 2006*). However, the number of cores present in muscle biopsies is not reflective of the degree of muscle weakness (*Wu et al., 2006*). Because proximal muscles of limbs are typically taken for muscle biopsies in patients, we examined the vastus lateralis and adductor magnus. Additionally, we examined the soleus muscle as it is commonly used in vitro studies of fatigue and RyR1-associated myopathy in mouse models. Analysis of these three muscles in 1-year old *Ryr1^AG/+* mice showed centrally located nuclei (13.2 ± 3.8 per 100 myofibers vs none detected in wild-type), core-like structures (*Table 1*), and decreased COX and NADH-TR staining, whereas no cores were observed in *Ryr1^+/+* muscle (*Figure 1D–I*). Focusing on type 1 fibers, core-like structures appear in 8.9 ± 1.8 per 100 NADH-TR stained type 1 fibers in 12-month old *Ryr1^AG/+* soleus muscle, which is comparable to the 12 cores per 100 NADH-TR stained fibers of 18-month old *Ryr1^IT* (*RyR1^I4895T*) soleus muscle, which has a congenital myopathy with cores (*Zvaritch et al., 2009*). In 2-month old *Ryr1^AG/+* muscle, we observed ultrastructural hallmarks of RyR1-associated myopathies (*Boncompagni et al., 2009*, *2010*), including z-line streaming (white arrows in *Figure 1J*: *Table 2*), sarcomeric degeneration, irregularly shaped mitochondria, core-like structures (white arrow in *Figure 1K*), and swelling of the t-tubules (*Figure 1L*) as well as internalized nuclei (9.8 ± 4.9 per 100 myofibers vs none detected in wild-type). The onset and extent of the pathological and histological changes in multiple muscle types are similar to a mouse CCD model, *RyR1^I4895T* (*Boncompagni et al., 2009*; *Zvaritch et al., 2009*; *Boncompagni et al., 2010*), altogether suggesting *Ryr1^AG* mice as an additional model for myopathies with core-like structures.

### RyR1^AG mutation causes a defect in calcium release

Ryr1 functions as a muscle isotype of ryanodine receptor that is mechanically activated by the dihydropyridine receptor to allow Ca$^{2+}$ release from the sarcoplasmic reticulum, thus evoking E–C coupling (*Fill and Copello, 2002*). To determine whether RyR1^AG mutant protein affects calcium release, we examined the effects of treatment with 4-chloro-*m*-cresol (4-C*m*C), a potent activator of RyR1 in skeletal muscle and RyR2 in cardiac muscle (*Zorzato et al., 1993*; *Choisy et al., 1999*). The domain, which is activated by 4-C*m*C, is located between residues 4007 and 4180 of RyR1 (*Fessenden et al., 2003*), near the *Ryr1^AG* mutation (E4242G). An increased sensitivity to 4-C*m*C has been linked to patients with malignant hyperthermia (*Herrmann-Frank et al., 1996a*; *Baur et al., 2000*) and ratiometric calcium imaging has been used to show hypersensitivity of 4-C*m*C in muscle cells from a patient with a mutation in RyR1 associated with malignant hyperthermia susceptibility (*Roesl et al., 2014*). Thus, we

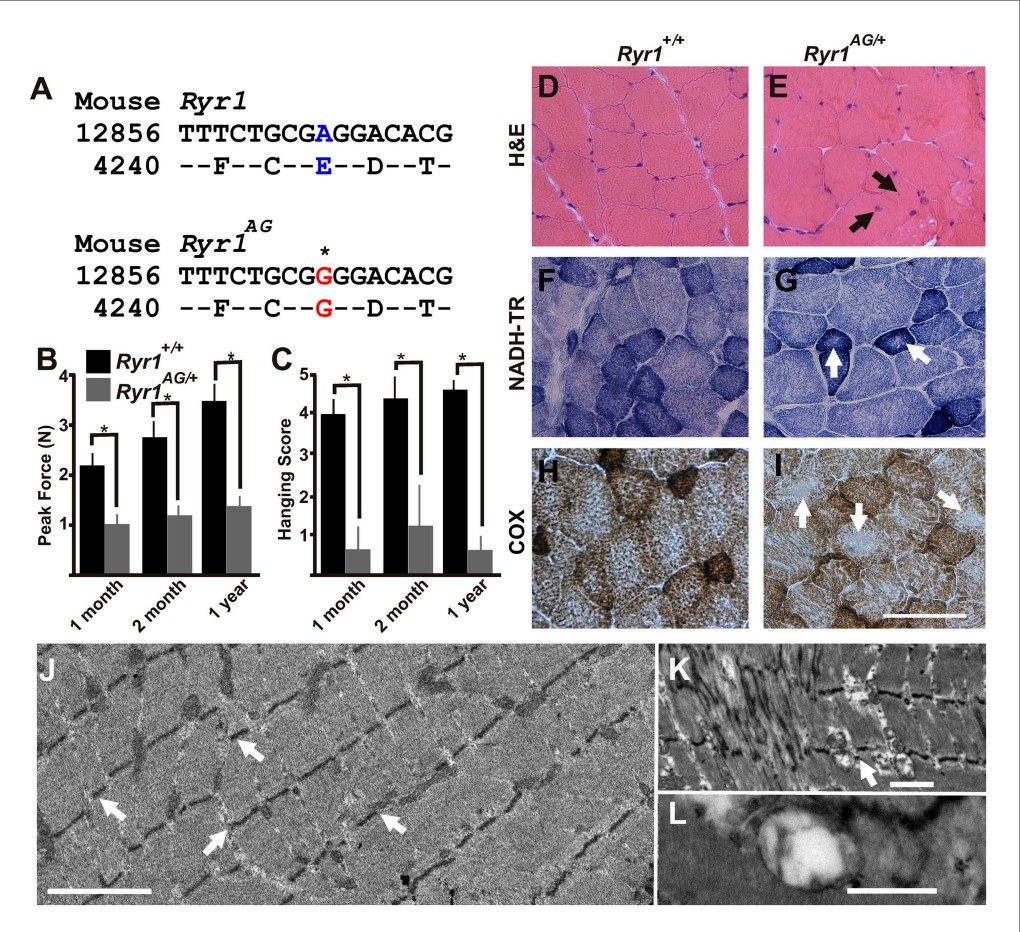

**Figure 1**. ENU-induced *Ryr1*[AG] mutation mimics clinical and pathological features of CCD in heterozygous mice. (**A**) Missense mutation in exon 93 of *Ryr1* changes A to G, resulting in substitution of glutamic acid with glycine (E4242G). (**B**) Average grip strength assayed using vertical digital push–pull strain gauge on *Ryr1*[+/+] and *Ryr1*[AG/+] mice at different ages raised on control 0.6% potassium diet. (**C**) In vivo hanging task determination of upper-body strength of *Ryr1*[+/+] and *Ryr1*[AG/+] mice at different ages (**B**, **C**, 10 trials/mouse, n = 5 per set). (**D**, **E**) H&E staining indicate central nuclei (arrows) in *Ryr1*[AG/+] (**E**) but not *Ryr1*[+/+] (**D**) vastus lateralis muscle from 1-year old mice raised on control 0.6% potassium diet. (**F**, **G**) NADH-TR staining indicates cores (white arrows) in 1-year old vastus lateralis of *RyR1*[AG/+] but not *Ryr1*[+/+]. (**H**, **I**) Cytochrome oxidase (COX) staining denotes a decrease in mitochondrial function (white arrows) in vastus lateralis muscle of *Ryr1*[AG/+] (**I**) compared to *Ryr1*[+/+] (**H**). Scale bar = 50 µm. (**J–L**) Transmission electron microscopy of 2-month old *Ryr1*[AG/+] soleus muscle. (**J**, **K**) Regions of Z line streaming and associated sarcoplasmic disruption and cores (white arrows) of myofibrils. (**L**) Enlarged T-tubules are present in type I fibers (scale bars: J = 1 µm; K = 2 µm; L = 0.2 µm ).

performed ratiometric $Ca^{2+}$ imaging using the calcium indicator, Fura-2 AM, and recorded the ratio of 340/380 nm fluorescence over a range of 4-C*m*C concentrations up to 1000 µM (*Zorzato et al., 1993*; *Herrmann-Frank et al., 1996b*). 4-C*m*C was applied to explanted soleus muscle from 2-month old *Ryr1*[AG/+] and littermate controls and ratiometric data of individual muscle fibers were normalized to the fluorescent ratiometric intensity at 1000 µM 4-C*m*C for each group. In wild-type mice, application of 200 µM 4-C*m*C induced measurable changes in Fura-2 signal (*Figure 2A*; 20 individual fibers from 4 mice). However, in littermate *Ryr1*[AG/+] muscle fibers, measurable changes in Fura-2 signal were not observed until the application of 600 µM 4-C*m*C (*Figure 2A*; 27 individual fibers from 5 mice). These data suggest that the *Ryr1*[AG/+] mutation results in decreased sensitivity to 4-C*m*C.

The reduced sensitivity of *Ryr1*[AG] to 4-C*m*C may suggest that less $Ca^{2+}$ is released into the myoplasm upon RyR1 activation. In myotubes expressing malignant hyperthermia RyR1 cDNAs, myoplasmic $Ca^{2+}$ is elevated due to a significant passive $Ca^{2+}$ leak from the SR (*Yang et al., 2007*). However, in

**Table 1.** Core-like structures in type 1 muscle fibers of 1-year old *RyR1*[AG/+] mice as revealed by NADH-TR and COX labeling

| | NADH-TR negative cores | COX negative cores |
|---|---|---|
| Vastus lateralis | 6.2 ± 2.4 | 7.8 ± 2.1 |
| Adductor magnus muscle | 9.3 ± 2.9 | 9.7 ± 2.2 |
| Soleus muscle | 8.9 ± 1.8 | 8.1 ± 1.4 |

Average values from 100 fibers with 10 slices per muscle and three muscles per group.

*Ryr1*[AG/+] muscle fibers, we found that myoplasmic $Ca^{2+}$ levels were significantly decreased compared to wild-type muscle fibers (*Figure 2B*; 85.3 ± 2.2% in *Ryr1*[AG/+] muscle fibers normalized to wild-type concentration). This suggests that a $Ca^{2+}$ leak that significantly elevates resting $Ca^{2+}$ is not present in *Ryr1*[AG/+] muscle fibers. In addition, the reduced sensitivity of *Ryr1*[AG/+] muscle fibers to 4-C*m*C may suggest an increase in SR $Ca^{2+}$. To examine SR calcium levels, 20 µM thapsigargin (TG), which inhibits SR $Ca^{2+}$ ATPases, was applied to Fura-2 labeled muscle fibers. Ratiometric analysis of $Ca^{2+}$ intensities provide evidence that *Ryr1*[AG/+] muscle fibers have an increase in TG-dependent SR $Ca^{2+}$ release (*Figure 2C,D*; 0.23 ± 0.05; 37 muscle fibers from 6 mice; p < 0.001) compared to wild-type muscle fibers (0.15 ± 0.03; 24 muscle fibers from 4 mice). This suggests increased levels of $Ca^{2+}$ in the SR of *Ryr1*[AG/+] muscle fibers and is consistent with the lack of a $Ca^{2+}$ leak and decreased RyR1-mediated $Ca^{2+}$ release from the SR in the mutant myofibers.

For muscle to function properly throughout the process of E–C coupling, the muscle fibers must maintain functional levels of $Ca^{2+}$ within the SR. In response to reduced myoplasmic $Ca^{2+}$, external $Ca^{2+}$ enters the myofiber to replenish internal stores through store-operated calcium entry (SOCE). To measure unidirectional ion flux through SOCE, we used $Mn^{2+}$ quenching of Fura-2 fluorescence in muscle fibers from *Ryr1*[AG/+] and wild-type mice as $Mn^{2+}$ has an increased affinity to Fura-2 compared to $Ca^{2+}$, thus decreasing $Ca^{2+}$-dependent myoplasmic fluorescence. To examine $Mn^{2+}$ quenching, Fura-2 fluorescence was measured in muscle fibers first washed in 0 mM $Ca^{2+}$, followed by $Mn^{2+}$, and finally Triton-X 100 with EGTA to provide a baseline measurement. *Ryr1*[AG/+] muscle fibers showed a substantially increased $Mn^{2+}$ entry rate (−0.08 ± 0.02 rate in wild-type; n = 16 muscle fibers in 4 mice) compared to that of wild-type (*Figure 2E,F*; −0.2 ± 0.02 rate in wild-type; n = 18 muscle fibers in 4 mice). These data suggest that $Ca^{2+}$ homeostasis is disrupted in *Ryr1*[AG/+] muscle.

## RyR1[AG] skeletal muscles show mitochondrial dysfunction

During excitation–contraction coupling, RyR1 floods $Ca^{2+}$ into the myoplasm for ATP-dependent contraction of muscle fibers. Immediately following contraction, $Ca^{2+}$ is pumped back into the SR and taken up by adjacent mitochondria for the purpose of ATP production. The decrease in myoplasmic calcium could be due to decreased release of $Ca^{2+}$ from the SR or increased uptake of $Ca^{2+}$ by the mitochondria, which prompted us to evaluate mitochondrial function. Local calcium release mechanisms or calcium 'sparks' communicate with adjacent mitochondria to spatially confine $Ca^{2+}$ release (*Rizzuto et al., 1993*; *Hajnoczky et al., 1995*; *Rizzuto et al., 1998*). In skeletal myotubes, RyR1 and RyR3 generate calcium sparks (*Ward et al., 2001*; *Weisleder et al., 2012*), with RyR1 being the predominant ryanodine receptor expressed in limb skeletal muscles (*Marks et al., 1989*; *Takeshima et al., 1989*; *Otsu et al., 1990*; *Zorzato et al., 1990*). In cardiac muscle, the in vivo depletion of RyR2 is sufficient to reduce mitochondrial $Ca^{2+}$ influx, oxidative metabolism, and ATP levels (*Bround et al., 2013*). Moreover, the activation of cytosolic $Ca^{2+}$ sparks can lead to the influx of mitochondrial calcium termed Rhod-2 labeled calcium 'marks' (*Pacher et al., 2002*). Rhod-2 distributes to mitochondria

**Table 2.** Number of Z line streaming sites within 100 µm² grid in 2-month old wild-type and *RyR1*[AG/+] soleus muscle

| Genotype | 12-month old | 2-month old | | | |
| | | 0.6% Diet | 5.2% Diet | Enalapril | Glibenclamide |
|---|---|---|---|---|---|
| *Ryr1*[+/+] | 0.48 ± 0.21 | 0.04 ± 0.01 | 0.03 ± 0.01 | 0.05 ± 0.03 | 0.09 ± 0.08 |
| *Ryr1*[AG/+] | 10.67 ± 0.82** | 3.75 ± 0.05** | 0.34 ± 0.16* | 1.13 ± 0.14** | 1.12 ± 0.02** |

Average values from five grids per mice and three mice per group. Asterisks indicate significant value *<0.05, **<0.005.

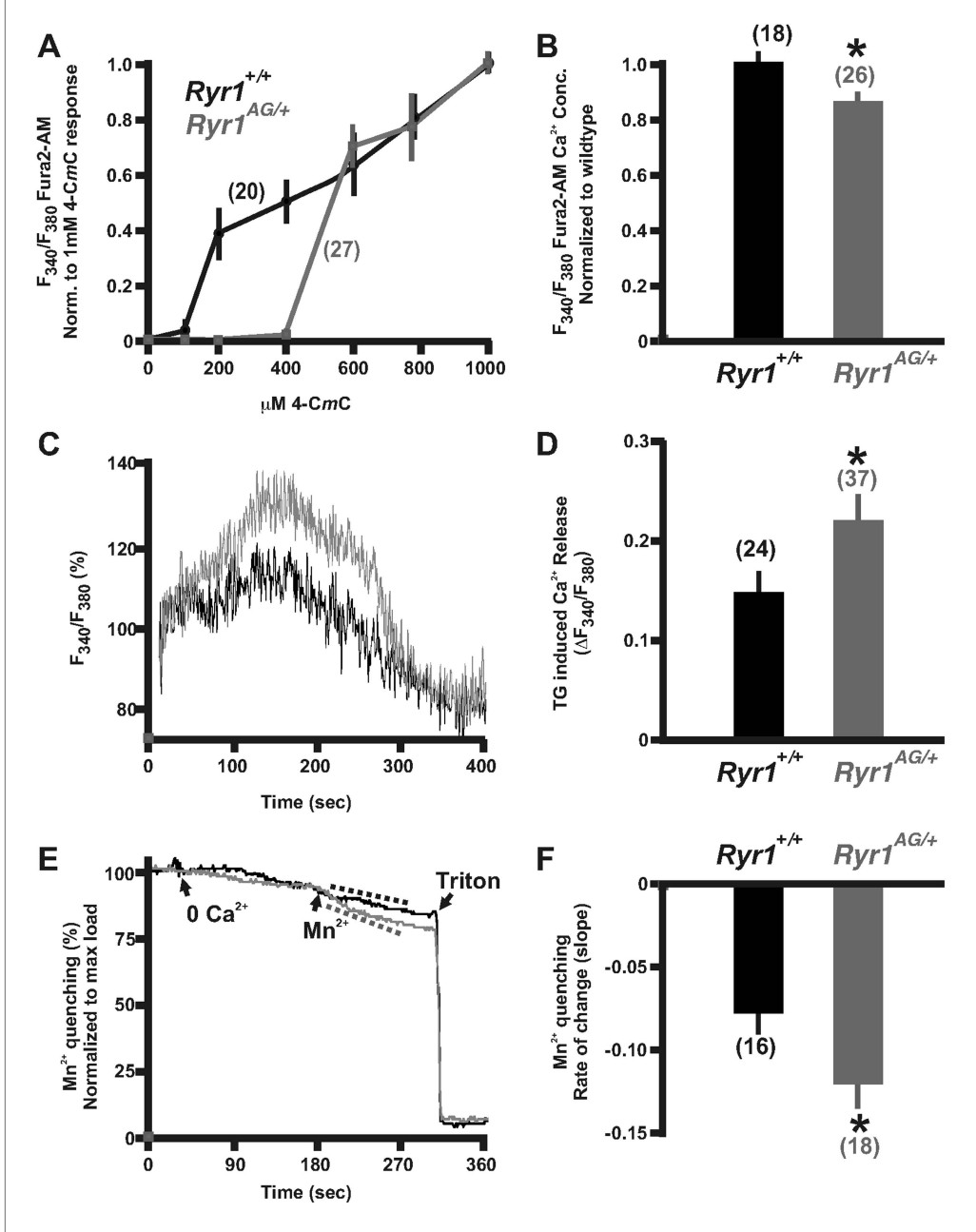

**Figure 2**. Calcium homeostasis is disrupted in $Ryr1^{AG/+}$ muscle. Fura-2 ratiometric imaging of myoplasmic $Ca^{2+}$ in 2-month old muscle. (**A**) Ratiometric analysis of 4-C*m*C sensitivity in soleus muscles of wild-type (black) and $Ryr1^{AG/+}$ littermates (grey). (**B**) Average ratiometric analysis of myoplasmic Fura-2 signal normalized to wild-type signal (n above bar represents number of analyzed fibers). (**C**) Representative fura-2 ratiometric signal of muscle fibers showing percent change after addition of 20 μM thapsigargin (TG) from wild-type (black) and $Ryr1^{AG/+}$ (grey) littermate. (**D**) TG induced maximum response in muscle fibers (n above bar represents number of analyzed fibers). (**E**, **F**) Representation and quantification of $Mn^{2+}$ quenching of Fura-2 fluorescence ratiometric signal illustrating increased influx of store operated calcium entry in soleus muscle fibers of $Ryr1^{AG/+}$ (grey) muscle fibers compared to wild-type (black) muscle fibers (dash lines represent slope). In (**E**), arrows indicate when media was introduced with 0 $Ca^{2+}$ followed by 0.5 mM $Mn^{2+}$.

when applied at low temperature and can be used as a measure of changes in fluorescence intensity (***Trollinger et al., 1997***). Influxes in mitochondrial calcium induce increased fluorescence intensities in Rhod-2 labeled mitochondria. In Ringer's solution, the number of mitochondria with changes in

fluorescence intensity over 10 min of imaging were significantly decreased in $Ryr1^{AG/+}$ compared to $Ryr1^{+/+}$ littermate isolated muscle fibers (6.1 ± 1.1 compared to 10.2 ± 1.4 per 1000 μm$^2$; *Figure 3D*), but the Rhod-2 intensity changes in mitochondria were increased after the application of 4-C*m*C to the bath in both $Ryr1^{AG/+}$ and $Ryr1^{+/+}$ isolated fibers (*Figure 3A–D*). Recent evidence in a mouse model with an induced mutation in RyR1 illustrates that leakage of SR Ca$^{2+}$ induces ROS activation and mitochondrial dysfunction (*Andersson et al., 2011*). However, no difference in MitoSOX, an indicator of superoxides,

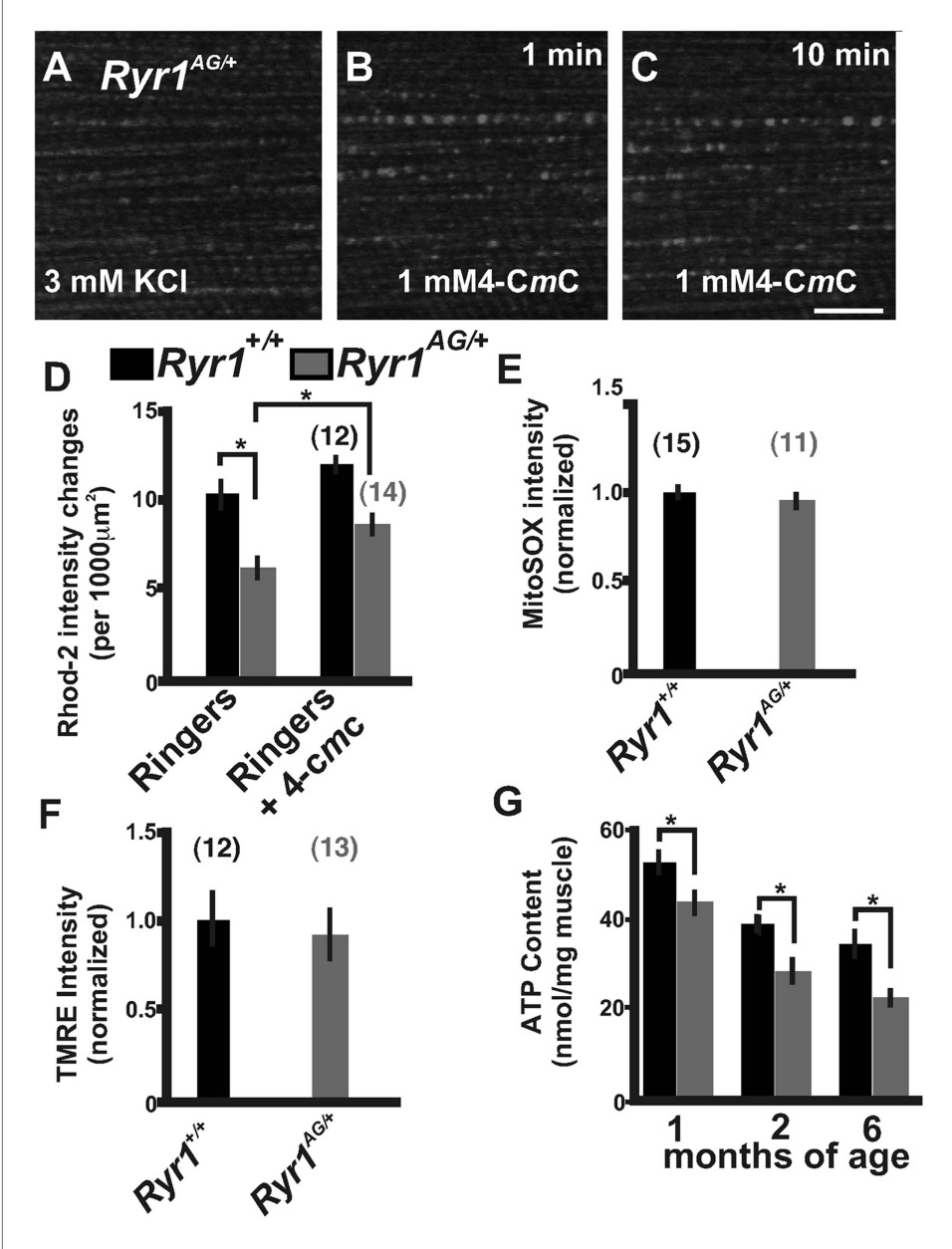

**Figure 3**. Defects in mitochondrial function in $Ryr1^{AG/+}$ mice. (**A–C**) Representative isolated fiber with Rhod2 fluorescence of 2-month old $Ryr1^{AG/+}$ in 3 mM KCl (**A**) and 1 min (**B**) and 10 min (**C**) after application of 1 mM 4-C*m*C (punctate fluorescence are calcium marks). (**D**) Quantification of mitochondrial calcium marks visualized with Rhod2-AM from isolated soleus muscle fibers from $Ryr1^{+/+}$ (black bars) and $Ryr1^{AG/+}$ (gray bars) mice. (**E**) Superoxide labeling of isolated muscle fibers from $Ryr1^{+/+}$ and $Ryr1^{AG/+}$ mice. (**F**) Intensity of TMRE labeling of isolated soleus muscle fibers from $Ryr1^{+/+}$ and $Ryr1^{AG/+}$ littermates with saponin. The numbers on top of bars in graphs represent number of fibers examined. (**G**) ATP content from $Ryr1^{+/+}$ and $Ryr1^{AG/+}$ isolated soleus muscle from age-matched mice (n = 4 for each age and group). Scale bar = 5 μm.

was detected in *Ryr1*[AG/+] compared to *Ryr1*[+/+] littermate 2-month old soleus muscle fibers (*Figure 3E*). This is consistent with our data showing that *Ryr1*[AG/+] muscle does not have an abnormal $Ca^{2+}$ leak.

The decrease in myoplasmic $Ca^{2+}$ and Rhod-2 labeled mitochondria $Ca^{2+}$ intensities, and the hyposensitivity in RyR1[AG/+] channels may lead to a reduction in mitochondrial generation of ATP. Mitochondrial membrane potential drives the generation of ATP. To analyze mitochondrial membrane potential, we used the mitochondrial membrane potential indicator TMRE along with the amphipathic glycoside, saponin, to avoid the potentially confounding influence of variations in plasma membrane potential. *Ryr1*[AG/+] isolated muscle fibers showed no significant difference relative to *Ryr1*[+/+] in overall TMRE labeling of mitochondria (*Figure 3F*). Therefore, muscle was examined for total muscle ATP generation. ATP determination showed ~30% less ATP in *Ryr1*[AG/+] soleus muscle compared to age-matched *Ryr1*[+/+] (1, 2, and 6-month old mice; n = 4 mice per group; *Figure 3G*). In human subjects that undergo fatigue, muscle fiber ATP levels can be reduced by up to 20% (*Sahlin et al., 1997*; *Karatzaferi et al., 2001a*, *2001b*; *Jones et al., 2009*), suggesting that the reduced ATP levels in *Ryr1*[AG/+] muscle might impair ATP-dependent enzyme activity during and after contraction (*Homsher, 1987*; *Clausen et al., 1991*; *Smith et al., 2005*; *Barclay et al., 2007*; *Clausen, 2013b*). These data suggest that dysfunctional $Ca^{2+}$ release in RyR1[AG] muscle may influence mitochondrial function and ATP production. Together the SR $Ca^{2+}$ release deficiencies and decreased mitochondrial ATP production may be sufficient to induce the myopathic phenotype in *RyR1*[AG/+] mice.

## Disruption of potassium homeostasis in *Ryr1*[AG/+] muscle

The significant depletion of ATP in *RyR1*[AG/+] muscle could consequently affect ATP-dependent mechanisms that normally act within the muscle, such as the influx of potassium to enhance membrane excitability. To determine whether potassium homeostasis is disrupted in *Ryr1*[AG/+] muscle, we used the potassium indicator PBFI-AM (see 'Materials and methods') in combination with ratiometric potassium imaging of muscle fibers. Experiments using wild-type soleus in normal Ringer's solution (3 mM $K^+$) showed that intracellular potassium concentration was 137 ± 7 mM (n = 23 fibers from 4 mice; *Figure 4A,C*). However, *Ryr1*[AG/+] muscle showed lower intracellular potassium concentrations of 106 ± 8 mM (n = 29 fibers from 4 mice; p < 0.005; *Figure 4B,C*), a similar decline as that observed for wild-type muscles undergoing fatigue (*McKenna et al., 2008*). Moreover, *Ryr1*[AG/+] muscle fibers showed a slow decrease in potassium fluorescent ratio intensity over the course of a minute suggesting an increase in $K^+$ ion permeability (*Figure 4D,E*; n = 23 fibers in 4 *Ryr1*[AG/+] mice and n = 29 fibers in 4 wild-type mice), which may be indicative of a potassium leak. We hypothesized that if *Ryr1*[AG/+] muscles have a defect in potassium homeostasis due to an increase in $K^+$ ion permeability, it might be possible to decrease the semi-permeability of the $K^+$ ion with the addition of external $K^+$. However, we did not observe a significant difference in serum $K^+$ levels in 2-month old *Ryr1*[AG/+] mice, but there was a 12 ± 3% increase in serum $K^+$ in 6-month old *Ryr1*[AG/+] mice relative to wild-type (*Figure 4—figure supplement 1*, n = 6 samples per group per age for *Ryr1*[+/+] and *Ryr1*[AG/+]). The acute addition of 7 mM KCl to the muscle increased PBFI-AM fluorescence intensity, suggesting inhibition of a potassium leak, while 0 mM KCl decreased PBFI-AM fluorescence intensity (*Figure 4D,E*; n = 23 fibers in 4 *Ryr1*[AG/+] mice and n = 29 fibers in 4 wild-type mice). Glibenclamide (2 μM) selectively binds to and inhibits SUR1/2 (*Porat et al., 2011*), subunits of the $K_{ATP}6.1$ and $K_{ATP}6.2$ channels in muscle (*Pedersen et al., 2009*). In wild-type adult mouse skeletal muscle, glibenclamide protects against fatigue caused by tetanic force (*Duty and Allen, 1995*). Additionally, in chick muscle fibers, glibenclamide increases the twitch and tetanus tension induced by application of caffeine, an agonist of RyR1 (*Andrade et al., 2011*). The acute addition of 2 μM glibenclamide to the 3 mM KCl bath resulted in increased intracellular $K^+$ in wild-type (n = 17 fibers in 4 mice) and *Ryr1*[AG/+] (n = 16 fibers in 4 mice) soleus muscle fibers (*Figure 4D,F*). Upon bath application of 7 mM KCl for 1.5 hr to examine a continuous counterbalancing of the $K^+$ leak, the intracellular potassium concentration in the mutant soleus increased to 132 ± 8 mM, similar to wild-type (*Figure 4G*, n = 25 fibers in 4 *Ryr1*[AG/+] mice and n = 26 fibers in 4 wild-type mice). Upon bath application of 2 μM glibenclamide for 1.5 hr, the intracellular potassium concentration in the mutant soleus increased to 138 ±12 mM, similar to wild-type (*Figure 4H*, n = 28 fibers in 4 *Ryr1*[AG/+] mice and n = 28 fibers in 4 wild-type mice). The decline in intracellular $K^+$ fluorescence in *Ryr1*[AG/+] muscle compared to wild-type muscle (*Figure 4A–E*) and the increase in intracellular $K^+$ upon $K_{ATP}$ channel inhibition (*Figure 4G,H*) suggest an increase in $K_{ATP}$ channel activity in *Ryr1*[AG/+] muscle. These data suggest that increased extracellular $K^+$ ion and inhibition of $K_{ATP}$ channels might prevent the excessive $K^+$ leak and protect the mutant muscle from decreased intracellular concentrations of $K^+$.

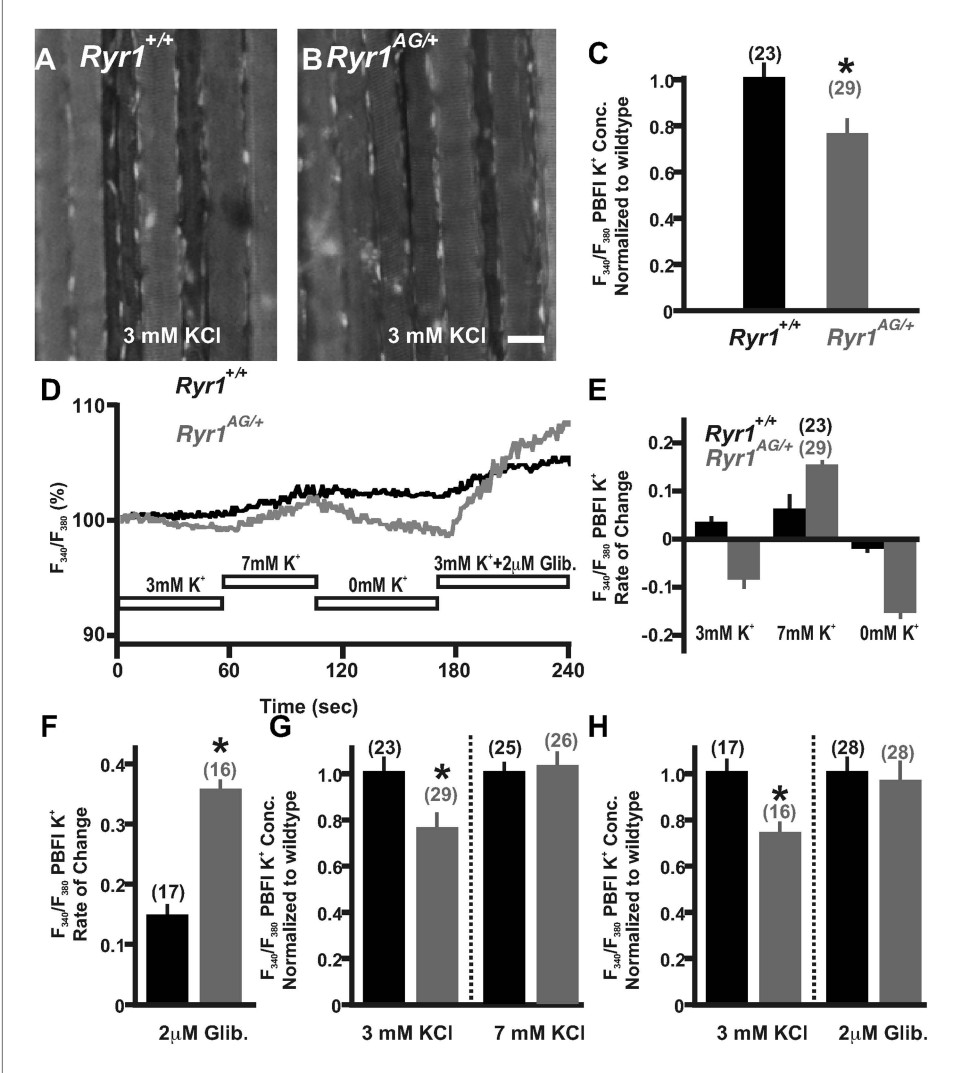

**Figure 4**. Detection and compensation of an internal potassium leak in *RyR1*[AG/+] muscle. Fluorescence imaging of PBFI at 340 nm in *Ryr1*[+/+] (**A**) and *Ryr1*[AG/+] (**B**) soleus muscle in 3 mM Ringer's solutions. (**C**) Ratiometric potassium imaging obtained at 340 and 380 nm wavelengths provided a ratio of fluorescence in *Ryr1*[+/+] (black) and *Ryr1*[AG/+] (grey) soleus muscle (normalized to *Ryr1*[+/+]). (**D**) Representation of the ratiometric imaging experimental paradigms used in *Figure 4* showing bath applications of 3 mM KCl, 7 mM KCl, 0 mM KCl, and 3 mM KCl with 2 µM glibenclamide in Ringer's solutions. (**E**) Slope of intracellular K[+] fluorescence intensities in experimental conditions. (**F, H**) Normalized intracellular K[+] concentration in *Ryr1*[+/+] (black) and *Ryr1*[AG/+] (grey) soleus muscle in 3 mM KCL (**F, H**) compared to soleus from contralateral limb in 7 mM KCl Ringer's solutions (**F**) or 3 mM KCL plus 2 µM Glibenclamide (**H**) (muscle was bathed in solutions for 1.5 hr before imaging, n is the number of fibers examined from four mice). (**G**) Rate of change in PBFI fluorescence after acute bath application of 2 µM Glibenclamide. Scale bar = 20 µm.

The following figure supplement is available for figure 4:

**Figure supplement 1**. Serum level measurements from *Ryr1*[+/+] (black bars) and *Ryr1*[AG/+] (grey bars) in 2- and 6-month old mice.

## Altered expression of genes and proteins involved in potassium homeostasis in *Ryr1*[AG/+] muscle

The decreased ATP concentrations and ratiometric potassium data in *Ryr1*[AG/+] muscle suggest that there may be an aberrant activation of mechanisms that regulate potassium efflux. Therefore, we examined several proteins involved in K[+] homeostasis in muscle of 2-month old *Ryr1*[AG/+] mice.

As mitochondrial dysfunction may confound the analysis of whole muscle, we prepared membrane-enriched lysates in which the plasma membrane and t-tubules were isolated from the fraction containing mitochondria, peroxisomes, and lysosomes (see 'Materials and methods'). The membrane-enriched preparations showed a 33 ± 6% increase in $K_{ATP}6.2$ protein in 2-month old $Ryr1^{AG/+}$ soleus muscle compared to $Ryr1^{+/+}$ littermates (*Figure 5A,B*; $p < 0.005$, the RNA encoding $K_{ATP}6.2$ (*Kcnj11*) was not altered in expression, *Table 3*, 0.6% diet), but no significant change in the protein expression of $K_{ATP}6.1$ or $K_{IR}2.1$. $K_{ATP}6.2$ channels localize to skeletal muscle membranes and t-tubules (*Banas et al., 2011*), suggesting that $K_{ATP}6.2$ channel function may be increased, leading to increased potassium efflux and disrupted potassium homeostasis, and contributing to the weakness in $Ryr1^{AG/+}$ muscle.

An increased number of $K_{ATP}$ channels in the membrane (or a change in activity) would be expected to alter current densities. In rat skeletal muscle, $K_{ATP}6.2$ pores have a single-channel conductance of around 80 pS, although the current densities (pA) of $K_{ATP}$ channels can vary between muscles (*Tricarico et al., 2006*). To examine current densities of $K_{ATP}$ channels in soleus muscle fibers, inside-out patch excision in ATP-free solutions was performed. Currents from excised patches of 2-month old wild-type soleus muscle fibers were −106 ± 12 pA with an average pipette resistance of 2.2 ± 0.4 MΩ after −60 mV voltage injection (n = 10 fibers from 3 mice). In 2-month old $Ryr1^{AG/+}$ soleus muscle fibers, the currents increased to −142 ± 24 pA with an average pipette resistance of 2.3 ± 0.3 MΩ (n = 10 fibers from 3 mice). Thus, $Ryr1^{AG/+}$ soleus muscle fibers showed a 34 ± 9% increase in normalized current densities compared to wild-type littermates (*Figure 5C,D*). Glibenclamide addition removed the current

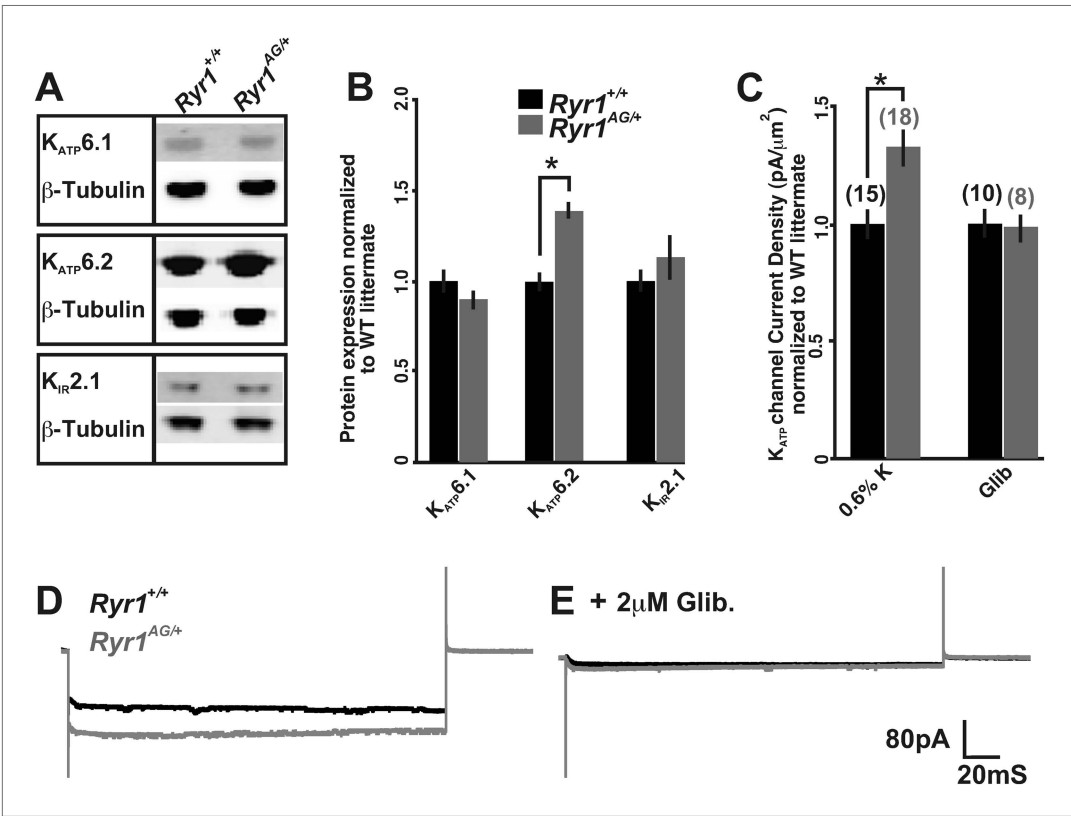

**Figure 5**. Altered activity of $K_{ATP}$ channels involved in potassium transport. (**A**) Western blots of membrane enriched lysate (that lacks mitochondria, peroxisomes, and lysosomes) from 2-month old wild-type and heterozygous $Ryr1^{AG/+}$ soleus muscle analyzed for $K_{ATP}6.1$, $K_{ATP}6.2$, and $K_{IR}2.1$. (**B**) Quantification of protein levels from Western blots of membrane enriched lysates. Each sample was first normalized to its own loading control and then the values from mutant and wild-type on the same blot were compared. Statistical analyses were determined from a minimum of 3 blots and at least two independent samples (n = 4 for each Western). (**C**) $K_{ATP}$ current densities of isolated soleus muscle fibers from 2-month old wild-type and $Ryr1^{AG/+}$ littermates with and without glibenclamide. Numbers on top of bars are number of fiber recordings. (**D–E**) Representative current density recordings with the same tip resistance from $Ryr1^{+/+}$ and $Ryr1^{AG/+}$ soleus muscle fibers (**E**) with and (**D**) without 2 μM glibenclamide.

**Table 3.** Quantitative RT-PCR of muscles from 2-month old wild-type and heterozygous $Ryr1^{AG/+}$ mice fed for 4 weeks on the indicated diets

| Gene | 0.6% K Diet | | 5.2% K Diet | |
|---|---|---|---|---|
| | $Ryr1^{+/+}$ | $Ryr1^{AG/+}$ | $Ryr1^{+/+}$ | $Ryr1^{AG/+}$ |
| *Abcc8* (SUR1) | | | | |
| Vastus lateralis | 1.00 ± 0.07 | 1.30 ± 0.21** | 1.05 ± 0.30 | 1.51 ± 0.33* |
| Tibialis anterior | 1.00 ± 0.04 | 1.20 ± 0.17*** | 1.09 ± 0.23 | 1.31 ± 0.27* |
| Adductor magnus | 1.00 ± 0.12 | 1.28 ± 0.22* | 0.98 ± 0.23 | 1.61 ± 0.42* |
| *Abcc9* (SUR2) | | | | |
| Vastus lateralis | 1.00 ± 0.19 | 0.81 ± 0.24 | 1.03 ± 0.20 | 1.09 ± 0.18 |
| Tibialis anterior | 1.00 ± 0.23 | 0.86 ± 0.21 | 0.99 ± 0.13 | 1.18 ± 0.20 |
| Adductor magnus | 1.00 ± 0.18 | 0.88 ± 0.19 | 1.01 ± 0.19 | 1.11 ± 0.11 |
| *Atpa1* (NKAα1) | | | | |
| Vastus lateralis | 1.00 ± 0.15 | 1.23 ± 0.21* | 0.94 ± 0.32 | 0.18 ± 0.06*** |
| Tibialis anterior | 1.00 ± 0.21 | 1.22 ± 0.13* | 0.75 ± 0.39 | 0.23 ± 0.06*** |
| Adductor magnus | 1.00 ± 0.26 | 1.25 ± 0.17* | 0.93 ± 0.29 | 0.24 ± 0.07*** |
| *Clc1* (CLC1) | | | | |
| Vastus lateralis | 1.00 ± 0.09 | 1.05 ± 0.19 | 0.90 ± 0.21 | 0.84 ± 0.25 |
| Tibialis anterior | 1.00 ± 0.10 | 1.10 ± 0.16 | 0.93 ± 0.11 | 0.82 ± 0.27 |
| Adductor magnus | 1.00 ± 0.12 | 1.03 ± 0.21 | 0.89 ± 0.15 | 0.88 ± 0.17 |
| *Kcnj2* ($K_{IR}$2.1) | | | | |
| Vastus lateralis | 1.00 ± 0.09 | 0.77 ± 0.12*** | 1.75 ± 0.17*** | 1.62 ± 0.13*** |
| Tibialis anterior | 1.00 ± 0.17 | 0.68 ± 0.17*** | 1.70 ± 0.26*** | 1.55 ± 0.15*** |
| Adductor magnus | 1.00 ± 0.18 | 0.59 ± 0.13*** | 1.67 ± 0.33** | 1.52 ± 0.18*** |
| *Kcnj8* ($K_{ATP}$6.1) | | | | |
| Vastus lateralis | 1.00 ± 0.20 | 0.56 ± 0.12*** | 0.97 ± 0.20 | 0.97 ± 0.23 |
| Tibialis anterior | 1.00 ± 0.13 | 0.74 ± 0.19*** | 0.87 ± 0.11 | 0.91 ± 0.12 |
| Adductor magnus | 1.00 ± 0.24 | 0.69 ± 0.16** | 1.16 ± 0.20 | 1.08 ± 0.17 |
| *Kcnj11* ($K_{ATP}$6.2) | | | | |
| Vastus lateralis | 1.00 ± 0.15 | 0.96 ± 0.17 | 1.10 ± 0.13 | 1.42 ± 0.14*** |
| Tibialis anterior | 1.00 ± 0.18 | 1.01 ± 0.16 | 1.05 ± 0.08 | 1.34 ± 0.09*** |
| Adductor magnus | 1.00 ± 0.17 | 0.94 ± 0.09 | 1.0 9± 0.06 | 1.38 ± 0.21*** |
| *Prkaa1* (AMPK) | | | | |
| Vastus lateralis | 1.00 ± 0.14 | 1.23 ± 0.15** | 1.11 ± 0.10 | 1.07 ± 0.14 |
| Tibialis anterior | 1.00 ± 0.04 | 1.11 ± 0.06** | 1.08 ± 0.05 | 1.07 ± 0.10 |
| Adductor magnus | 1.00 ± 0.11 | 1.20 ± 0.11*** | 1.01 ± 0.11 | 1.10 ± 0.06 |

Values normalized to GAPDH before normalization to $Ryr1^{+/+}$ control on 0.6% diet. Asterisks indicates significant value * = 0.05, ** = 0.001, *** = 0.005.

differences between $Ryr1^{+/+}$ and $Ryr1^{AG/+}$ muscle, suggesting that the channels opened in the inside-out patch were $K_{ATP}$ channels (**Figure 5C,E**, n = 8 fibers from three mice for both $Ryr1^{+/+}$ and $Ryr1^{AG/+}$). These data indicate an increase in $K_{ATP}$ channel activity in the membrane of $Ryr1^{AG/+}$ muscle and this could explain the increased potassium efflux in spite of the decreased ATP levels in mutant muscle.

Potassium transport activity of $K_{ATP}$ channels is positively regulated by the AMP-activated protein kinase AMPK (*Prkaa1*) (**Wang et al., 2005**), and this transcript was upregulated in 2-month old $Ryr1^{AG/+}$ muscles (**Table 3**, 0.6% diet). The sulfonylurea receptors (SUR) are ATP-binding cassette (ABC) transporters and subunits of $K_{ATP}$ channels. SUR1 (*Abcc8*), which binds to $K_{ATP}$6.2, was significantly

upregulated in $Ryr1^{AG/+}$ muscles at the RNA level (**Table 3**). Thus, the increased expression of genes that encode proteins upstream of $K_{ATP}6.2$ channels or that interact with $K_{ATP}6.2$ channels might contribute to the increased activity of $K_{ATP}6.2$ channels in membrane and t-tubules. ClC-1-dependent chloride conductance is increased during muscle fatigue (**Pedersen et al., 2009**), yet ClC-1 transcripts were not significantly altered in $Ryr1^{AG/+}$ muscle compared to $Ryr1^{+/+}$ littermates. $Ryr1^{AG/+}$ muscle showed a rapid increase in intracellular potassium when $K_{ATP}$ channels were blocked (**Figure 4C,D,F**), suggesting that mechanisms of $K^+$ transport into the cell are active in these muscles but may not be able to overcome the potassium leak and compensate for decreased intracellular $K^+$. $Na^+$, $K^+$-ATPase activity maintains $Na^+$ and $K^+$ equilibrium at the membrane in the muscle (**Kristensen and Juel, 2010**). Transcripts of the pump subunit $Na^+$, $K^+$-ATPase α1 were increased in $Ryr1^{AG/+}$ muscle (**Table 3**). Taken together, these data suggest that expression levels of genes and proteins in the potassium transport pathway are dysregulated when RyR1-dependent calcium release is disrupted.

## Amelioration of myopathic phenotype through potassium supplementation

As our in vitro studies showed that increased extracellular $K^+$ leads to a normalization of intracellular $K^+$ in $Ryr1^{AG/+}$ muscles, we then asked whether increased serum $K^+$ could alter the in vivo phenotype of $Ryr1^{AG/+}$ mice. Serum potassium levels can be increased through diet (**Christensen et al., 2010**) or with the angiotensin-converting enzyme (ACE) inhibitor, enalapril (**Cleland et al., 1985**). Therefore, we asked whether these treatments might decrease $K^+$ ion permeability in muscle and whether this could ameliorate the physiological and pathological symptoms of the myopathy in $Ryr1^{AG/+}$ mice. To test these hypotheses, $Ryr1^{AG/+}$ and $Ryr1^{+/+}$ mice were weaned onto 0.6% $K^+$ (level of potassium in standard mouse chow), and four weeks later, when muscle weakness is already observed (**Figure 1B,C**), they were given either 5.2% $K^+$ diet or enalapril (0.02 mg/ml in drinking water) for an additional four weeks (**Figure 6—figure supplement 1**, for blood pressure readings). Examination of soleus muscle fibers from these mice showed that internal potassium concentrations were increased in $Ryr1^{AG/+}$ soleus fibers, similar to wild-type, following 5.2% $K^+$ diet (145 ± 22 mM compared to 147 ± 20 mM; **Figure 6A**) or with enalapril (147 ± 18 mM compared to 148 ± 25 mM; **Figure 6A**). As noted in **Figure 1**, weakness in 2-month old $Ryr1^{AG/+}$ mice on 0.6% $K^+$ diets was clearly evident with an ~50% reduction in grip strength and inability to hang onto the wire, but in line with our hypothesis, $Ryr1^{AG/+}$ mice treated with enalapril showed a significant improvement in the wire hanging task (**Figure 6B,C**). Most dramatically, the 5.2% $K^+$ diet rescued muscle strength in $Ryr1^{AG/+}$ mice (**Figure 6B,C**). Thus, potassium supplementation and an FDA-approved drug can rescue muscle weakness in mice carrying the $Ryr1^{AG}$ mutation.

Histological examination is the hallmark diagnosis of RyR1-related myopathies. On control 0.6% $K^+$ diets, internally located nuclei and decreased mitochondrial activity were detected in $Ryr1^{AG/+}$ muscle (**Figure 6D,E–H**). Strikingly, muscle from $Ryr1^{AG/+}$ mice on high-potassium diet resembled $RyR1^{+/+}$ muscle (**Figure 6D,I–L**), while enalapril treatment showed a less dramatic improvement in muscle histology (**Figure 6D,M–P**). The number of Z line streaming areas was also significantly reduced by these therapies (**Table 2**). Thus, even after observable muscle dysfunction, myopathic pathology can be rescued with increased extracellular potassium.

Chronic treatment of $Ryr1^{AG/+}$ animals for 1 month with a 5.2% potassium diet returned $K_{ATP}6.2$ protein expression to levels similar to those seen in diet-treated $Ryr1^{+/+}$ littermates (**Figure 7A**). We also noted that protein levels of the inwardly rectifying potassium channel $K_{IR}2.1$ were increased by 64 ± 23% in $Ryr1^{AG/+}$ muscle following 5.2% $K^+$ diet (**Figure 7A**, p < 0.01), suggesting that increased $K_{IR}2.1$ channel activity may also assist in rectifying the $K^+$ ion leak. This physiological rescue and the normalization of $K_{ATP}$ expression suggest a decrease in $K_{ATP}$ channel activity. $K_{ATP}$ current density recordings in $Ryr1^{AG/+}$ soleus muscle fibers following 5.2% potassium diet were returned to levels similar to wild-type on normal diet (−107 ± 16 pA in 2.0 ± 0.3 MΩ), whereas currents in $Ryr1^{+/+}$ on 5.2% diet were substantially decreased (−74 ± 15 pA in 2.1 ± 0.4 MΩ; **Figure 7B**). ATP levels were also increased by 5.2% potassium diet in $Ryr1^{AG/+}$ soleus muscle, similar to $Ryr1^{+/+}$ (38.5 ± 6.2 and 41.1 ±8.3 nmol ATP/mg soleus, respectively on 5.2% diet; **Figure 7C**). Moreover, the 5.2% $K^+$ diet increased mitochondrial calcium marks in isolated fibers from $Ryr1^{AG/+}$ soleus muscles to a rate similar to wild-type littermates, with no effect on ROS activity (**Figure 7D,E**). These data indicate a rescue of the myopathic muscle physiology by diet, indicating potential therapeutic avenues for RyR1-related myopathies in human.

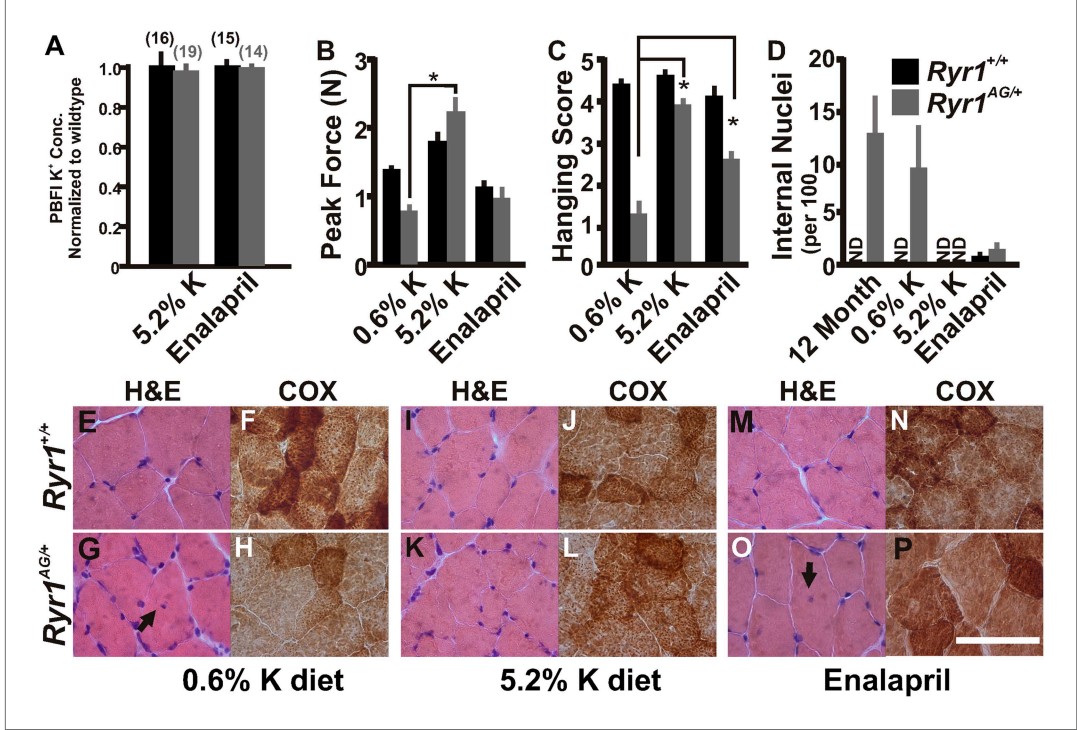

**Figure 6**. Increased potassium diet can rescue muscle strength and reverse the CCD histology and myopathy. (**A**) Normalized internal potassium concentrations of soleus muscle from 2-month old $Ryr1^{+/+}$ and $Ryr1^{AG/+}$ mice fed 0.6% K+ diet for 4 weeks, then placed on 5.2% K+ diet or 0.6% diet + enalapril for 4 weeks, and then bath exposed to different extracellular potassium concentrations. (**B**, **C**) Average grip strength (**B**, five trials/mouse and 5 mice/set; p values <0.001) and in vivo hanging task (**C**, 10 trials/mouse, n = 5 per set; p values <0.001) assayed from 2-month old $Ryr1^{+/+}$ (black bar) and $Ryr1^{AG/+}$ (grey bar) mice maintained for 4 weeks on control 0.6% K+ diet, 5.2% K+ diet, or 0.6% K+ diet supplemented with enalapril. (**D**) Quantification of number of internalized nuclei per 100 myofibers in 12-month old mice or in 2-month old mice maintained for 4 weeks on control 0.6% K+ diet, 5.2% K+ diet, or 0.6% K+ diet supplemented with enalapril (n = 10 per muscle, n = 3 per set of muscles for total of n = 30; p values <0.001). (**E**–**P**) Vastus lateralis myofibers from 2-month old $Ryr1^{+/+}$ and $Ryr1^{AG/+}$ mice maintained for 4 weeks on control 0.6% K+ diet, 5.2% K+ diet, or 0.6% K+ diet supplemented with enalapril. Cross-sections stained with H&E (left panels) and COX (right panels). $Ryr1^{AG/+}$ mice on 5.2% K diet show increased COX staining and no internalized nuclei, similar to $Ryr1^{+/+}$. These pathological features are still observed in $Ryr1^{AG/+}$ mice on 0.6% K+ diets. Enalapril increases COX staining but some internalized nuclei are observed, even in $Ryr1^{+/+}$. Scale bar = 50 μm; error bars as standard error of the mean (SEM).

The following figure supplement is available for figure 6:

**Figure supplement 1**. Blood pressure measurements from $Ryr1^{+/+}$ (black bars) and $Ryr1^{AG/+}$ (grey bars) in 2-month old mice.

## Amelioration of myopathic phenotype through glibenclamide

Glibenclamide can inhibit $K_{ATP}$ channels, which are overexpressed in the mutant muscle (**Figure 5A**), and rectify the K+ ion leak when acutely applied in vitro (**Figure 4C**). To extend these findings in vivo, we asked whether this FDA-approved drug could ameliorate the myopathic pathology in mice. To test this, glibenclamide (15 mg/kg/day) was orally administered to 1-month old mice for 30 days. $Ryr1^{AG/+}$ mice treated with glibenclamide showed significantly increased grip strength (**Figure 8A,B**). ATP content and mitochondrial calcium changes in intensities in glibenclamide treated $Ryr1^{AG/+}$ mice were increased to similar levels as $Ryr1^{+/+}$ mice (36.4 ± 3.9 and 39.3 ± 4.3 nmol ATP/mg soleus muscle, respectively; n = 4 mice per condition, **Figure 8C**; 11.8 ± 1.5 and 11.1 ± 1.8 mitochondrial changes in intensities, respectively; n = 8 muscle fibers for each group, **Figure 8D**) with no alteration in MitoSOX labeling (data not shown). $Ryr1^{+/+}$ and $Ryr1^{AG/+}$ mice on glibenclamide therapy showed no significant difference in internal potassium concentrations (140 ± 9 mM in $Ryr1^{+/+}$ soleus fibers compared to

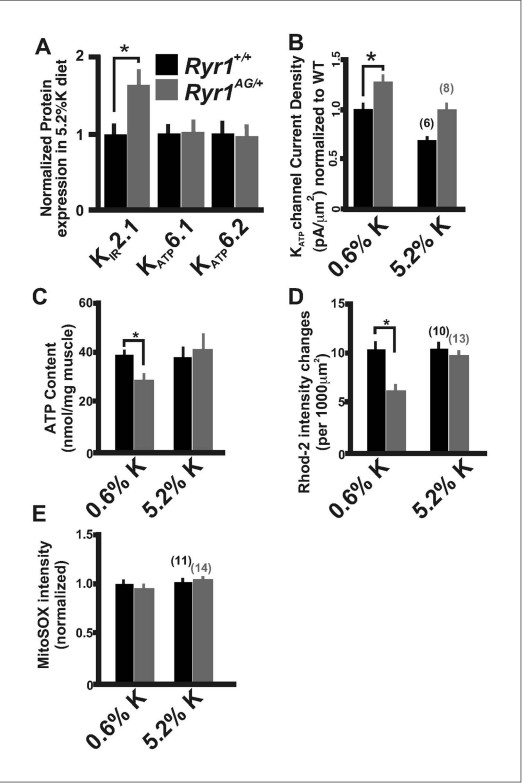

**Figure 7**. Increased potassium diet influences $K_{ATP}$ channel activity and mitochondrial function. (**A**) Membrane enriched lysate (lacking mitochondria, peroxisomes, and lysosomes) from soleus muscle of 2-month old $Ryr1^{+/+}$ and $Ryr1^{AG/+}$ mice fed 5.2% potassium supplemented diet for 4 weeks analyzed by Western blot for $K_{ATP}6.1$, $K_{ATP}6.2$, and $K_{IR}2.1$ (* is p < 0.001). (**B**) $K_{ATP}$ current densities of isolated soleus muscle fibers from $Ryr1^{+/+}$ and $Ryr1^{AG/+}$ littermates after 4 weeks on 5.2% K diet compared to 0.6% K diet. (**C**) ATP content from $Ryr1^{+/+}$ and $Ryr1^{AG/+}$ isolated soleus muscle from mice on 0.6% or 5.2% potassium diet. (**D**) Quantification of calcium marks visualized with Rhod2-AM from isolated soleus muscle from $Ryr1^{+/+}$ and $Ryr1^{AG/+}$ mice with and without diet therapy. (**E**) Superoxide labeling of isolated soleus muscle from $Ryr1^{+/+}$ and $Ryr1^{AG/+}$ mice with and without therapy. Error bars as standard error of the mean (SEM).

145 ± 28 mM in $Ryr1^{AG/+}$ soleus fibers; n = 17 and 20 muscle fibers, respectively; normalized in *Figure 8E*). Current densities from $Ryr1^{AG/+}$ soleus muscle fibers from mice treated with glibenclamide were −82 ± 13 pA with an average pipette resistance of 2.1 ± 0.3 MΩ (n = 8 fibers from 4 muscles; *Figure 8F*) and showed no significant difference from wild-type littermates on glibenclamide (−93 ± 18 pA with 2.2 ± 4 MΩ tip resistance, n = 8 fibers from 4 muscles; *Figure 8F*). Furthermore, a significant decrease in internal nuclei was detected in $Ryr1^{AG/+}$ muscles (9.8 ± 5.2 in 0.6% K$^+$ diet compared to 4.8 ± 2.7 in glibenclamide diet; n = 4 mice per condition; *Figure 8G*). Administration of glibenclamide also rescued the histological manifestations of the myopathy in $Ryr1^{AG/+}$ mice (*Figure 8H–K*, *Table 2*), suggesting an alternative treatment with an FDA-approved drug that does not involve modulation of serum potassium levels. Together our in vivo data suggest that the mitochondrial dysfunction in $Ryr1^{AG/+}$ soleus muscle is significantly improved through 5.2% potassium diets and glibenclamide administration. These studies suggest that rescue of the weakened state of muscles in this mouse myopathic model are, in part, due to a potassium-dependent mechanism.

## High-potassium diet affects biomarkers for potassium homeostasis

The levels of protein and gene expression might serve as measurements of recovery in $Ryr1^{AG/+}$ mice following administration of a high-potassium diet. Indeed, as shown in *Figure 7A* and *Table 3*, we found that expression of proteins or transcripts that encode $K_{ATP}6.1$, $K_{ATP}6.2$, AMPK, and SUR2 were returned to normal levels on high-potassium diet in $Ryr1^{AG/+}$ adult muscle, whereas Na$^+$, K$^+$-ATPase α1 transcripts were considerably decreased and $K_{IR}2.1$ protein was increased in high-potassium diet in $Ryr1^{AG/+}$ muscle. Thus, potassium supplementation rescues the expression of key molecules required for potassium homeostasis in $RyR1^{AG/+}$ muscle, suggesting that

$K_{IR}2.1$, $K_{ATP}6.1$, $K_{ATP}6.2$, *Prkaa1*, *Abbc9*, and Na$^+$, K$^+$-ATPase α1 are potential biomarkers for myopathies associated with mutations in RyR1 and for evaluation of future potential therapies.

## CCD patient muscles show upregulation of potassium homeostasis biomarkers

Central core disease is the most common diagnosed myopathy due to mutations in RyR1 (*Jungbluth, 2007*). Using the molecular insights gained from our myopathic mouse model, we examined these biomarkers in RNA isolated from human skeletal muscles collected from CCD patients and controls without muscle disease (normalized to a pooled control muscle sample, see methods). Expression levels of potential biomarkers from human muscle samples of healthy individuals, patients with RyR1 mutations causing CCD, and patients with CCD without identified RyR1 mutations were compared

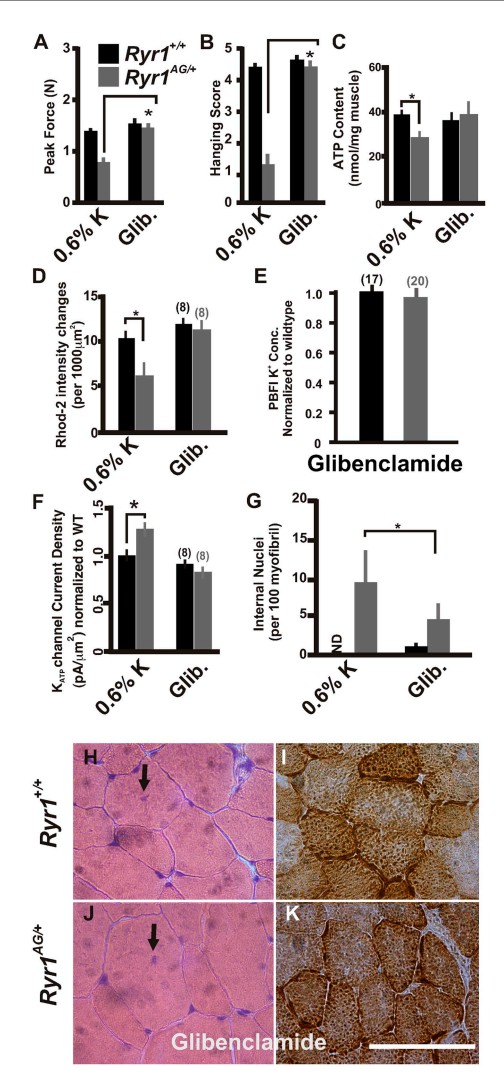

**Figure 8.** Inhibition of $K_{ATP}$ channels can reverse the histological and myopathic phenotypes. (**A**, **B**) Average grip strength (**A**, five trials/mouse and 5 mice/set; p values <0.001) and in vivo hanging task (**B**, 10 trials/ mouse, n = 5 per set; p values <0.001) assayed from 2-month old $Ryr1^{+/+}$ (black bar) and $Ryr1^{AG/+}$ (grey bar) mice maintained for 4 weeks on 0.6% K diet and glibenclamide (15 mg/kg/day). (**C**) ATP content from soleus muscle of $Ryr1^{+/+}$ and $Ryr1^{AG/+}$ mice after 4 weeks on 0.6% K diet without or with glibenclamide (n = 4 muscles per condition). (**D**) Quantification of calcium marks visualized with Rhod2-AM from isolated soleus muscle from $Ryr1^{+/+}$ and $Ryr1^{AG/+}$ mice with and without glibenclamide therapy. (**E**) Normalized internal potassium concentrations after glibenclamide therapy. (**F**) $K_{ATP}$ current densities of isolated soleus muscle fibers from $Ryr1^{+/+}$ and $Ryr1^{AG/+}$ littermates after 4 weeks on glibenclamide therapy compared to 0.6% K diet. (**G**) Quantification of number of internalized nuclei per 100 myofibers in all conditions (n = 10 per muscle, n = 3 per set of muscles for total of n = 30; p values <0.001).
*Figure 8. Continued on next page*

(*Table 4*; *Figure 9*). While *PRKAA1* was increased in all muscle biopsies compared to the pooled human RNA sample, excluding it as a useful biomarker, all CCD patients showed a significant increase in expression of *ABCC8* (SUR1), *KCNJ2* ($K_{IR}2.1$), *KCNJ8* ($K_{ATP}6.1$), and *KCNJ11* ($K_{ATP}6.2$), indicating that these four transcripts may be useful biomarkers for CCD in humans. Together, these data suggest specific biomarkers that could aid in diagnosis and treatment of CCD.

## Discussion

In this study, we present a myopathic mouse model with core-like pathology that illuminates mechanisms underlying muscle weakness and mitochondria dysfunction. First, we identified a mutation in *RyR1* that induces a myopathy with ultrastructural hallmarks of cores similar to the $Ryr1^{I4895T/+}$ mutant mouse, a known mouse model of CCD (*Zvaritch et al., 2007, 2009*; *Boncompagni, 2010*). Second, we used this myopathic mouse model to identify a novel link between decreased internal calcium (*Figure 2*), decreased mitochondrial ATP production (*Figure 3*), and altered potassium homeostasis (*Figure 4*), mechanisms that are important in controlling excitability in muscle. Third, we provide evidence that $K_{ATP}$ channels are elevated in the membrane and overactive, leading to increased potassium efflux in this myopathic mouse model, which could have direct effects on membrane excitability. Fourth, we provide clinically relevant findings that suggest a potassium-enriched diet, perhaps in combination with the sulfonylurea class of anti-diabetic drugs, which are potential therapies for myopathies with defects in potassium homeostasis. Using multiple assays and different muscles with different fiber type compositions, our data are consistent and show muscle weakness, pathological changes, and gene expression changes. The consistency of our data between muscle types further strengthens our hypotheses and conclusions. It should be noted that malignant hyperthermia, a disease associated with *RyR1* mutations, is sensitive to potassium chloride (*Moulds and Denborough, 1974*), and therefore a testing method in patients or biopsy samples need to be initiated prior to potassium supplementation. Importantly, our work identifies potential diagnostic biomarkers of disease in patients with CCD and potentially non-RyR1 myopathies, which could be utilized to identify patients who may be candidates for potassium modulation-based therapies.

The physiological mechanisms underlying RyR1-related myopathy are not well defined. One histological hallmark of RyR1-related myopathy is

*Figure 8. Continued*

(**H–K**) Vastus lateralis myofiber from *Ryr1*[+/+] and *Ryr1*[AG/+] mice maintained for 4 weeks on glibenclamide. Arrow in *Ryr1*[+/+] myofiber (F) shows the rare occurrence of internal nuclei. Cross-sections stained with H&E (left panels) and COX (right panels). *Ryr1*[AG/+] mice on glibenclamide show increased COX staining, similar to *Ryr1*[+/+]. Scale bar = 50 µm; error bars as standard error of the mean (SEM).

the presence of muscle cell cores, which indicate focal mitochondrial loss and decreased ATP production (*Boncompagni et al., 2009*, *2010*; *Jungbluth et al., 2011*). The *Ryr1*[AG/+] mutant mouse muscles have decreased intracellular calcium, core-like structures and decreased ATP content, as well as other ultrastructural hallmarks of core-like diseases. Data from our model suggest that proper calcium release from internal stores is required for mitochondrial production of ATP and increased ATP levels, which in a feedback loop can then stimulate RyR1-dependent calcium release (*Meissner et al., 1986*; *Laver et al., 2001*). As muscle ATP decreases, $K_{ATP}$ channels open to efflux $K^+$ from the myoplasm to the interstitial space, which may protect against excessive muscle contraction. Efflux of $K^+$ into the t-tubules depolarizes the tubular membranes and inhibits the opening of sodium channels (*Fraser et al., 2011*). Sodium channel inhibition decreases the probability that a nerve-derived action potential will activate DHPR receptors that mechanically activate Ryanodine receptors, and thus myoplasmic release of stored $Ca^{2+}$ is reduced. Therefore, in the *Ryr1*[AG/+] model, RyR1 dysfunction, decreased internal $Ca^{2+}$ release, potassium leakage and increased $K_{ATP}$-dependent efflux may potentiate muscle weakness and muscle deterioration associated with core-like structures. The chronic low-level calcium uptake into the mitochondria may then induce long-term mitochondrial dysfunction. The histological appearance of cores in the center of the muscle but more normal staining near the plasma membrane may be explained by the possibility that more centralized mitochondria receive less calcium, whereas SOCE and calcium influx from external sources at the plasma membrane may provide buffering to mitochondria that are adjacent to the plasma membrane.

Our data suggest that the SR calcium release deficiencies and decreased mitochondrial ATP production might be sufficient to induce the myopathic phenotype in *RyR1*[AG/+] mice. ATP provides the energy to cause muscle contraction, and relative ATP/ADP levels are important in providing information about muscle fatigue and ion homeostasis to the cell. In skeletal muscle, 90% of ATP is used for muscle contractions by the myosin ATPase, $Ca^{2+}$ ATPase, and $Na^+/K^+$ ATPase (*Homsher, 1987*; *Clausen et al., 1991*; *Smith et al., 2005*; *Barclay et al., 2007*; *Clausen, 2013b*). Additional ATP-dependent enzymes may be particularly sensitive to a 30% loss of ATP in *RyR1*[AG/+] muscle and this might contribute to a significant effect on excitability and membrane homeostasis after muscle contraction and

**Table 4.** Relative variation of quantitative RT-PCR of control RNA and RNA from human muscle biopsies

| Patient | RYR1 Mutation | Congenital Myopathy | ABCC8 (SUR1) | ATPA1 (NKAα1) | KCNJ8 ($K_{ATP}$6.1) | KCNJ11 ($K_{ATP}$6.2) | PRKAA1 (AMPK) |
|---|---|---|---|---|---|---|---|
| 1 | No | No | 0.98 ± 0.02 | 1.35 ± 0.05 | 2.15 ± 0.19 | 1.59 ± 0.39 | 3.88 ± 0.06 |
| 2 | No | No | 0.94 ± 0.08 | 0.85 ± 01 | 1.06 ± 0.06 | 0.78 ± 0.16 | 4.35 ± 0.02 |
| 3 | Yes | Yes | 13.61 ± 0.07 | 4.35 ± 0.11 | 4.95 ± 0.05 | 3.28 ± 0.16 | 2.90 ± 0.11 |
| 4 | Yes | Yes | 30.68 ± 0.12 | 9.40 ± 0.08 | 7.72 ± 0/04 | 3.12 ± 0.18 | 15.45 ± 0.13 |
| 5 | Yes | Yes | 19.37 ± 0.34 | 19.98 ± 0.54 | 6.48 ± 0.07 | 3.92 ± 0.09 | 17.85 ± 0.09 |
| 6 | Yes | Yes | 108.61 ± 1.88 | 41.43 ± 0.31 | 9.74 ± 0.08 | 16.24 ± 0.89 | 17.32 ± 0.11 |
| 7 | Yes | Yes | 4.69 ± 0.07 | 37.72 ± 0.36 | 17.45 ± 0.14 | 5.21 ± 0.35 | 4.04 ± 0.08 |
| 8 | Yes | Yes | 46.10 ± 0.22 | 3.64 ± 0.04 | 22.96 ± 11 | 7.77 ± 17 | 10.73 ± 0.09 |
| 9 | Yes | Yes | 159.20 ± 1.07 | 9.61 ± 0.16 | 67.76 ± 0.24 | 8.33 ± 0.42 | 35.59 ± 0.29 |
| 10 | No | Yes | 34.18 ± 0.19 | 1.09 ± 0.11 | 9.64 ± 0.13 | 4.43 ± 0.29 | 8.23 ± 0.02 |
| 11 | No | Yes | 22.30 ± 0.66 | 1.09 ± 0.10 | 45.14 ± 0.11 | 13.82 ± 0.81 | 21.03 ± 0.04 |

Values normalized to *GAPDH* before normalization to pooled control human muscle RNA. Red indicates significant value 0.005; Blue indicates significant value 0.05.

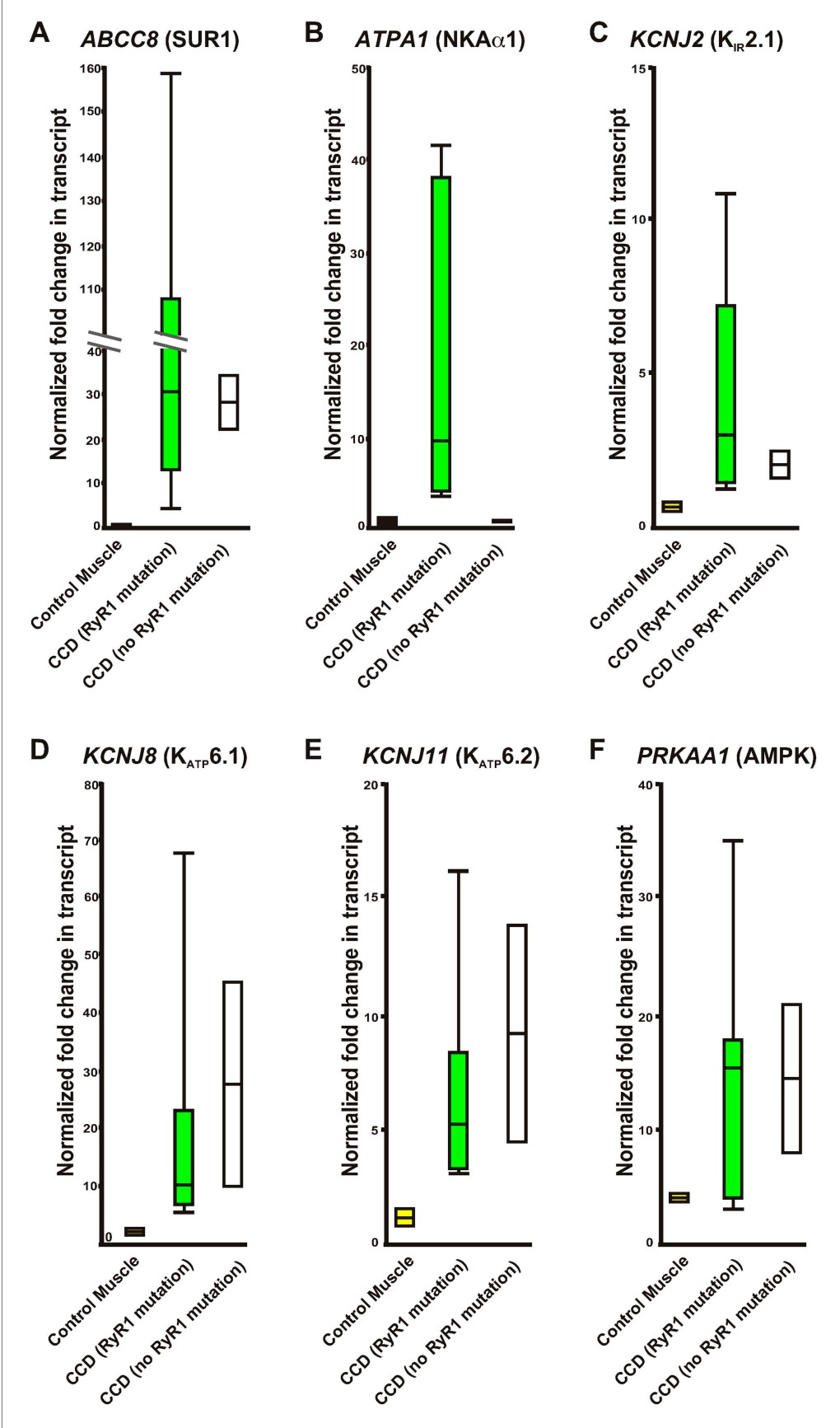

Figure 9. Continued on next page

*Figure 9. Continued*

**Figure 9**. Relative variation of quantitative qRT-PCR of human control RNA and RNA from human muscle biopsies. Whisker plots derived from the data in *Table 2* of control muscle (yellow), congenital myopathy (CCD) with a mutation in RyR1 (green), and without a mutation in RyR1 (white). (**A**) *ABCC8*, (**B**) *ATPA1*, (**C**) *KCNJ2*, (**D**) *KCNJ8*, (**E**) *KCNJ11*, (**F**) *PRKAA1*. Values normalized to *GAPDH* before normalization to pooled control human muscle RNA.

fatigue. During skeletal muscle fatigue and fluctuations in potassium homeostasis, $K_{ATP}$ channels open and efflux potassium from the muscle (*Pedersen et al., 2009*). Moreover, plasma membrane and t-tubule $K_{ATP}$ channels become more active as ATP is utilized and ADP accumulates (*Baczko et al., 2005*). $K_{ATP}$ channels are mechanistically linked to the DHPR-RyR1 complex (*Tricarico et al., 1999*) but the ATP binding sites within RyR1 are at a considerable distance from the point mutation in the RyR1$^{AG}$ protein (*Popova et al., 2012*). Finally, $K_{ATP}$ channel activity increases potassium conductance within the t-tubules after an action potential (*Gramolini and Renaud, 1997*; *Cifelli et al., 2008*; *Banas et al., 2011*). These data suggest a mechanistic link between E–C coupling, RyR1 function, ATP levels, $K_{ATP}$ channel function, and potassium homeostasis.

In our mouse model of a myopathy with core-like structures, increased intake of dietary potassium increases interstitial K+, which leads to an increase in myoplasmic K+. The resulting increase in interstitial K+ might counterbalance the K+ leak permeability and increase membrane excitability. A more excitable membrane increases the likelihood of voltage-dependent DHPR-RyR1 release of calcium from the SR during an action potential. Increased calcium in the myoplasm will enhance mitochondrial uptake of calcium. We propose that the waves of increased calcium into the mitochondria enhance ATP production (*Griffiths and Rutter, 2009*), which inhibits $K_{ATP}$ channel opening and reverses the K+ leak, and stimulates RyR1-dependent calcium release from internal stores (*Meissner et al., 1986*; *Laver et al., 2001*), further perpetuating mitochondrial ATP production in this model of myopathy in which the phenotype can be ameliorated by increased potassium. However, we cannot rule out the possibility that the potassium supplementation may work indirectly at the muscle by depolarizing the locomotor circuit to increase neurotransmitter release at the neuromuscular junction and potentiate the voltage-dependent DHPR-RyR1 release of calcium.

Defective potassium mechanisms appear to be the hallmark of *Ryr1$^{AG/+}$* mice as evidenced by 1) increased $K_{ATP}$ channel current that would drive K+ efflux, 2) decreased internal K+ concentrations, and 3) a potassium leak. Altogether this may suggest an increase in interstitial K+, but serum K+ levels may not correlate with interstitial K+ levels (*Clausen, 2013a*) and increased interstitial K+ would be expected to be rapidly removed by diffusion and through the capillary network of blood vessels. We did not observe a significant difference in serum K+ levels in 2-month old *Ryr1$^{AG/+}$* mice, but there was an increase in serum K+ in 6-month old *Ryr1$^{AG/+}$* mice relative to wild-type (*Figure 4—figure supplement 1*). In vitro, the rapid rise in intracellular K+ in *Ryr1$^{AG/+}$* muscle bathed in 3 mM K+ and treated with glibenclamide, an inhibitor of $K_{ATP}$ channels, is in part due to the transport of K+ into the myoplasm through $K_{IR}2.1$ and Na+, K+-ATPases. These mechanisms that transport K+ into the myoplasm in *Ryr1$^{AG/+}$* muscles may not be sufficient in vivo to compensate for the K+ leakage or the increased activation of $K_{ATP}$ channels located on the plasma membrane. We suggest that *Ryr1$^{AG/+}$* myopathic muscles have a K+ leak that persistently decreases intracellular K+ and leads to a loss of membrane excitability.

Amelioration of muscle weakness through low-cost potassium supplementation could have profound impact on patients suffering from core myopathies and these data may have important implications for other disease states. Recent guidelines from the World Health Organization suggest that the world population is consuming inadequate levels of dietary potassium, which can lead to heart disease, stroke, respiratory distress, and increased fatigue (*Hartl, 2013*). Our studies suggest that *Ryr1$^{AG/+}$* muscle is in a myopathic state due to decreased intracellular Ca$^{2+}$ levels, decreased K+ levels, decreased ATP levels and increased activation of $K_{ATP}6.2$ channels, which may mimic a fatigue-like state. Currently, myopathies with mutations in RyR1 are not considered metabolic myopathies, even though RyR1-S2844D mice with an intracellular calcium leak and muscle weakness exhibit decreased levels of mitochondrial activity and metabolic enzymes (*Andersson et al., 2011*). Considering that our data suggest mitochondrial dysregulation of metabolic pathways and potassium signaling in *Ryr1$^{AG/+}$* muscle and muscles from patients with CCD, we were intrigued by the findings that some fatigue-inducing metabolic myopathies can be rescued through dietary potassium supplementation

(*Villabona et al., 1987*; *Caradonna et al., 1992*; *Ismail et al., 2001*). In the future, it will be of interest to determine whether the positive response to increased potassium levels is applicable to myopathies due to other RyR1 mutations and to other genetic causes. Our data also suggest that muscles of human CCD patients are sensitive to pathways that mediate potassium homeostasis. *ATPA1* ($Na^+$, $K^+$-ATPase α1) showed increased expression only in CCD patients with mutations in *RYR1*, suggesting a selective sensitivity to this ATP-sensitive transporter. With our suggestion that $K_{ATP}6.2$ channel inhibition may help protect skeletal muscles from the formation of cores, one might predict that the human patient data would show consistently higher levels of KCNJ11 ($K_{ATP}6.2$) transcripts in the muscle biopsies. This is the case in CCD patients, although the level is variable. This variability could be due to the location of the muscle biopsy, as $K_{ATP}6.2$ channels in mice are more highly expressed in glycolytic muscle than oxidative skeletal muscle fibers (*Banas et al., 2011*). These biomarkers may therefore provide a less invasive approach to identify patients with CCD myopathies and may eventually identify patients who are suitable candidates for treatment of CCD through modulation of potassium signaling pathways.

## Materials and methods

### Forward genetic screen and mouse strains
ENU mutagenesis was performed as described (*Kasarskis et al., 1998*) on males of C57BL/6J background and then outcrossed onto 129S1/Svlmj background to score G3 embryos at embryonic day 18.5 for recessive mutations that affect embryonic locomotion. The *Ryr1^m1Nisw* allele showed muscle weakness when heterozygous (described here) and a non-motile embryonic phenotype when homozygous (unpublished results). The gene mutation was mapped using data from homozygous mutants, starting with a panel of 96 MIT and SKI SSLP markers, which mapped the Line1–4 mutation to the proximal third of chromosome 7. The genetic region containing the mutation was narrowed to 3 Mb by the use of additional MIT SSLP markers on chromosome 7 and further meiotic mapping which followed linkage between the phenotype and C57BL/6J markers. As Ryr1 was a strong candidate gene in the critical interval, a complementation cross was performed between *Ryr1^AG* (*Ryr1^m1Nisw*) and *Ryr1 ^tm1TAlle* (*dyspedic* null allele generated by Richard Allen, [*Buck et al., 1997*]). These alleles did not complement as E18.5 trans-heterozygotes showed a similar non-motile phenotype as homozygotes of either allele. To identify the mutation in Line1-4, genomic DNA in overlapping segments of the *Ryr1* gene was amplified by PCR from phenotypic E18.5 embryos and compared with control E18.5 C57Bl/6J DNA. Subsequently, embryos were genotyped as follows: tissue was placed in tail lysis buffer (100 mM Tris–Cl, pH 8.0, 5 mM EDTA, 0.2% SDS, 200 mM NaCl) overnight. DNA was amplified using Taqman Gold (Applied Biosystems) with primer set to *RyR1* mutation (forward primer: GTGGAGTGTGGTGCTGTATATC forward and CCGTCACTGTCACCTTCTTG reverse primers). PCR amplification was performed for 35 cycles at 55°C. PCR product was sent to Barbara Davis Center Molecular Biology Service Center at the University of Colorado Denver for sequencing. For non-sequencing genotyping, the MIT SSLP markers, D7MIT114 and D7MIT267, were amplified to examine linkage between phenotypic background (C57BL/6J) and control background (129/SvJ). Transcript accession for RyR1 was NM_0091009 and the ENU-induced change was A12864G (protein NP_033135, resulting in E4242G). All of the data presented here were obtained after outcrossing >8 generations onto 129S1/Svlmj background.

### Electron microscopy
Soleus muscles of 2-month old mice were fixed with 3.5% glutaraldehyde in 0.1 M cacodylate buffer (pH 7.2) at 37°C. The central region of muscle was dissected to maximize the endplate regions, followed by a secondary post-fixation in 2% osmium tetroxide + 1.5% potassium ferrocyanide in 0.1 M cacodylate buffer for 1 hr at room temperature. Samples were dehydrated though an ethanol series (50, 70, 90, 95, and 100% ETOH) and embedded in Epon/Araldite resin polymerized overnight at 60°C. Ultrathin sections were cut, placed on copper mesh grids, and double contrasted with 2% aqueous uranyl acetate and Reynold's lead citrate. Grids were photographed using a FEI Technai G2 BioTwin transmission electron microscope.

### Histochemistry
COX and NADH staining protocols are available at html://neuromuscular.wustl.edu/pathol/histol. Histostained samples were examined on a Zeiss LSM 510 META laser scanning microscope.

## [Ca²⁺]i/PBFI measurements

Soleus muscle was dissected from limb with tendons attached and bifurcated at the proximal tendon. Bifurcated muscle was affixed to glass bottom culture dishes (MatTeK) with Vetbond (3M, St. Paul, MN) and cleaned of debris and adipose tissue. Tyrode buffer: 140 mM NaCl, 5 mM KCl, 10 mM HEPES, 2 mM MgCl2, 2.5 mM CaCl2, pH 7.2, 290 mosm. The ratiometric cell-permeant calcium indicator Fura-2 AM (Molecular Probes, Eugene, OR, USA) was administered to the preparation for 45 min at 33°C then washed two times for 20 min at room temperature. Muscles were analyzed at 33°C using the 340/380 filters in a Zeiss Axio Observer Z1 with a Photometrics CoolSNAP HQ2 camera for wide-field imaging, then performing ratio intensity analysis. To deplete SR $Ca^{2+}$ store, soleus fibers were treated with 10 μm thapsigargin for 2–4 min to induce $Ca^{2+}$ release. The perfusion solution was then switched to 0.5 mM $Mn^{2+}$ for 2–4 min to observe the extent of $Mn^{2+}$ entry and establish the rate of SOCE. For all measurements of SOCE by $Mn^{2+}$ quenching, the fluorescence signal was normalized using values determined by lysis of the cells with a solution containing 0 mM $Ca^{2+}$/EGTA and 0.1% Triton X-100 at the end of the experiment. All experiments were conducted at ~33°C.

## [K⁺]i/PBFI measurements

For [K+]i measurements, soleus muscle was affixed to glass bottom culture dishes (MatTeK, Ashland, MA) with Vetbond (3 M). The ratiometric cell-permeant potassium indicator PBFI-AM (5 μM; Life Technologies, Grand Island, NY) together with 0.2% Pluronic F-127 for enhanced dye loading was loaded for 30 min in Ringer's solution (3 mM K+). After loading, muscles were washed for 30 min to remove excess dye and to allow de-esterification of the AM dye. Data were collected from 10 regions per muscle with three muscles per condition. Muscles were analyzed at 33°C using the 340/380 filters in a Zeiss Axio Observer Z1 with a Photometrics CoolSNAP HQ2 camera for wide-field imaging, then performing ratio intensity analysis. To calibrate intracellular potassium concentrations, PBFI ratio intensities were fitted to PBFI intensities due to the variable $K^+$ solutions with the addition of 10 μM gramicidin in extracellular solutions and using the intensity of ~10 fibers of interest per muscle in >3 muscles from each condition. Calibration solutions were prepared with appropriate volumes of a high [K+] solution with potassium gluconate.

## Western analysis

Muscle tissue from the limbs was dissected and washed in PBS and resuspended in RIPA buffer supplemented with Complete Mini Protease Inhibitor Cocktail (Roche Basel, Switzerland) and Phosphatase Inhibitor Cocktail Set II (EMD Millipore, Billerica, MA). Samples were sonicated for 1 min (15 s on, 15 s off, 25% power) and incubated at 4°C for 1 hr on a rocking platform. Protein was quantified with the Bio-Rad Protein Assay (BioRad, Hercules, CA). All samples were incubated at 37°C for 30 min prior to loading. Protein was separated on 4–12% Bis-Tris gels (Invitrogen) and transferred to Immobilon-FL PVDF membranes (Millipore). Membranes were blocked in 5% dry milk in TBST for 1 hr. Primary antibody incubations were performed overnight at 4°C while rocking. Antibodies were diluted in TBST supplemented with 5% wt/vol BSA (Sigma, St Louis, MO). Secondary antibody incubations were performed for 1 hr at room temperature in TBST supplemented with 5% dry milk. Western blots were imaged using the Odyssey Infrared Imaging System (Li-Cor, Lincoln, NE). Quantitation was performed using Odyssey v3.0 software (Li-Cor). Primary and secondary antibodies were used as follows: KIR6.1 ($K_{ATP}$6.1; Santa Cruz, goat, 1:200), KIR6.2 ($K_{ATP}$6.2; Abcam, goat, 1:1000), KIR2.1 (Abcam, rabbit, 1:1000), Goat anti-Rabbit IRDye 680 (Li-Cor, 1:10,000), Goat anti-Mouse IRDye 800CW (Li-Cor, 1:10,000), Donkey anti-Goat IRDye 680LT (Li-Cor, 1:30,000), Donkey anti-Mouse 800CW (Li-Cor, 1:10,000).

## Membrane extract lysate

Mitochondrial and cytosolic protein fractions were isolated using differential centrifugation. Immediately after sacrifice, soleus skeletal muscles from *RyR1*[+/+] and *RyR1*[AG/+] mice were extracted, rinsed in saline solution, weighed, then cut into four fragments, and homogenized in 1:5 wt/vol ice-cold isolation buffer containing 0.21 M mannitol, 0.07 M sucrose, 0.005 M HEPES, 0.001 M EDTA, 0.2% fatty acid-free BSA, pH 7.4, using glass homogenizer. The homogenate was centrifuged at 600×*g* for 10 min at 4°C. The supernatant was then centrifuged at 12,000×*g* for 20 min. After the second spin, the supernatant (membrane, t-tubule and cytosol) was separated from the pellet (peroxisome, lysome, and mitochondria) and was stored at −80°C until used for Western blot assays.

## ATP content

Soleus skeletal muscles were assayed immediately after homogenization using buffer described in membrane extract lysate to determine ATP content with a luciferin–luciferase based bioluminescence assay. The methodology of the ATP determination kit is provided in the experimental protocol by Molecular Probes (A-22066, Eugene, OR, USA). In order to determine ATP content, freshly extracted homogenate was added to a cuvette containing reaction buffer, D-luciferin, luciferase, and DTT and placed in a Sirius luminometer v.2.2 (Berthold Detection Systems, Pforzheim, Germany). Known concentrations of ATP standards were used to establish a standard curve.

## Quantitative RT-PCR

RNA was extracted from muscles using Trizol followed by DNaseI digestion and clean-up (Qiagen RNAeasy Minikit, Venlo, Limburg) and reverse transcribed using random hexamer primers and SuperScript III Reverse Transcriptase (Invitrogen) and amplified using TaqMan Universal PCR Master Mix (Applied Biosystems). Quantitative PCR was performed on a Roche LightCycler 480 Real-Time PCR System. Calculations were performed by a relative standard curve method. Probes for target genes were from TaqMan Assay-on-Demand kits (Applied Biosystems). Samples were adjusted for total RNA content by GAPDH in mice on diets and for the human samples. Control human muscle RNA was purchased from Amsbio (R1234171-50) and Invitrogen (AM7982). The RNA samples were compared to a five sample human RNA skeletal muscle pool (R1234171-P). RNA samples of myopathies were generously provided by Telethon Biobank (Italy) and University of Colorado Neurology Department following University of Colorado COMIRB approval for human subject research.

## Potassium diets and drug treatments

Diets were based on studies examining hypertension in mice (*Grimm et al., 2009*). Diets were purchased through Harlan Laboratories, Inc. and are TD.10942 (0.6% K$^+$) and TD.94121 (5.2% K$^+$). These diets have similar ratios of three K$^+$ sources. Following weaning (1 month of age), animals were placed on potassium diets *ad libitum* for 4 weeks followed by determination of muscle strength and molecular and histological examination. For enalapril and glibenclamide studies, following weaning, mice were placed on 0.6% K+ diet and given enalapril in drinking water (0.02 mg/ml) or glibenclamide via daily oral gavage (15 mg/kg/day) for 4 weeks.

## Muscle fiber isolation

Skeletal muscle fibers were isolated from the soleus muscles by a classical enzymatic dissociation process; muscles were incubated for 1 hr at 37°C in Tyrode's solution containing collagenase (2.0 mg/ml, Sigma, Type 1) for patch-clamp experiments or 1.5 hr at 37°C in Tyrode's solution containing collagenase (2.2 mg/ml, Sigma, Type 1) for calcium imaging and mitochondrial function experiments.

## Patch-clamp experiments

After enzyme treatment, muscles were rinsed with Tyrode's solution and kept in low calcium Ringer's solution at 4°C until analysis. Ringer's solution used during collection of isolated muscle fibers contained 145 mM NaCl, 5 mM KCl, 1 mM MgCl$_2$, 0.5 mM CaCl$_2$, 5 mM glucose, and 5 mM HEPES and was adjusted to pH 7.2. The patch-pipette solutions contained 150 mM KCl, 2 mM CaCl$_2$, and 1 mM HEPES (pH 7.2). The bath solution contained 150 mM KCl, 5 mM EGTA, and 1 mM HEPES (pH 7.2).

Intact skeletal muscle fibers were separated from the muscle mass by gently triturating the muscle with a plastic 1000 µl pipette. Experiments were performed in inside-out configurations by using the standard patch-clamp technique. Initial pipette resistance was between 1.3–3.0 MΩ. Channel currents were recorded during voltage steps going from 0 mV of holding potential to −60 mV voltage membrane ($V_m$) immediately after excision, at 20–22°C, in the presence of KCl on both sides of membrane patches. Useful patches required minimal seal resistance of 1.1 GΩ. The mean currents were calculated by subtracting the baseline level from the open-channel level of each current trace. A minimum of three traces was collected for each seal. Digital averaging of traces was performed using CLAMPFIT (Molecular Devices, Sunnyvale, CA). ATP (5 mM) or glibenclamide (1 mM) was used to examine the pharmacological response of the channel.

## Grip strength and Hanging Wire Task

To evaluate muscle weakness in vivo, we compared *RyR1$^{AG/+}$* and *RyR1$^{+/+}$* mice on the 5.2% K$^+$ or 0.6% K$^+$ diet without or with glibenclamide or enalapril using grip strength tests (*Costa et al., 2010*) and

wire hanging task (*Ogura et al., 2001*). Similar tests have been used to evaluate muscle weakness in CCD mice (*Loy et al., 2011*). Performance in Hanging Wire Task was scored on a 0 to 5 scale according to *Loy et al. (2011)*: 0, immediately fell off the bar; 1, hung onto bar with two forepaws; 2, hung onto bar with two forepaws and attempted to climb onto the bar; 3, hung onto the bar with two forepaws and one or both hind paws; 4, hung onto the bar with all four paws and tail wrapped around the bar; 5, hung onto the bar with all four paws and tail wrapped around the bar and escaped onto one of the supports. Body weight was measured before and after therapies with no significant difference in weight (data not shown).

## Malignant hyperthermia and isoflurane sensitivity

Malignant hyperthermia test was performed by placing *Ryr1*$^{+/+}$ and *Ryr1*$^{AG/+}$ mice in a 41°C humidified incubator for 30 min. To test for sensitivity to halogenated anesthetics, stage 3 anesthetic isoflurane (~$5.5 \times 10^{-5}$ ml/cm$^3$) was administered through a nasal cone to *Ryr1*$^{AG/+}$ mice at a maximum of 30 min exposure.

## Mitochondrial functionality

Isolated soleus muscle fibers from 2-month old *Ryr1*$^{+/+}$ and *Ryr1*$^{AG/+}$ were examined for mitochondrial function. MitoTracker Deep Red FM (Invitrogen) was added to all fibers to identify the mitochondria and Powerload (Invitrogen) was added to calcium indicators for 20 min. To examine mitochondria calcium changes in intensity, Rhod2-AM (Invitrogen) was added to the fibers in Normal Ringer's solution for 30 min at room temperature and then the cultures were washed with normal Ringers. To examine mitochondrial membrane potential, TMRE (Invitrogen) with saponin (Sigma) were added at the manufacturer's recommended dose for 30 min at room temperature. FCCP (carbonyl cyanide 4-(trifluoromethoxy)phenylhydrazone; ABCAM, Cambridge, England) was added at the end of the experiment to eliminate the mitochondrial membrane potential and the value was used to normalized to the experimental fluorescent TMRE value. To examine superoxides, MitoSOX (Invitrogen) indicator was added at the manufacturer's recommended dose for 15 min at 37°C. Antimycin A (Sigma) was used as a positive control at the end of the experiment to normalize to the experimental signal. We placed the fiber cultures into a Zeiss Environmental incubation system (5% humidity, 95% O$_2$) at 22°C. Images of mitochondrial regions (3 per fiber and 5 fibers per muscle in 3 muscles) were taken on a Zeiss LSM510 laser-scanning microscope of 1000 μm$^2$ fields. Scans were taken every 2 min. All data were normalized to baseline per mitochondrial region to give a Δf/f value (Image J).

## Statistical analysis

The unpaired two-tailed Student's t test was used to compare means for statistical differences. Data in the manuscript are represented as mean ± SEM unless otherwise indicated. $p < 0.05$ was considered significant.

## Serum analysis

Normal methods to collect serum do not provide an accurate representation of interstitial (extracellular) potassium levels (*Mazzaccara et al., 2008*). However, to determine whether any overt discrepancies of serum potassium, sodium, and chloride can be detected, serum was collected from 2-month and 6-month old wild-type and *Ryr1*$^{AG/+}$ mice. Analysis was performed using a Starlyte ISE Analyzer (Alfa Wassermann, West Caldwell, NJ).

## Study approval

All experiments were conducted in accordance with the protocols described in the Guide for the Care and Use of Laboratory Animals (NIH. Revised, 2011) and we received University of Colorado COMIRB approval for human subject research.

## Acknowledgements

We thank Lori Bulwith, Radu Moldovan, and Chris Rivald for technical assistance, Alberto Costa for providing the apparatus to measure muscle strength, and Emily Bates, William Betz, and John Caldwell for comments on the manuscript. We thank Telethon Genetic Network of Biobanks and UC Denver Neurology Department for the use of RNA samples of muscle biopsies from CCD patients. Angela Ribera supplied physiology equipment (NIH R01 NS25217). This work was supported by the Department of Pediatrics and by the Neuroscience Program NS 48154. MGH was supported by

postdoctoral fellowships from MDA69338 and NIH F32 NS059267. LN was an investigator of the Howard Hughes Medical Institute. The authors have declared that no conflict of interest exists.

## Additional information

### Funding

| Funder | Grant reference number | Author |
| --- | --- | --- |
| Howard Hughes Medical Institute | | M Gartz Hanson, Jonathan J Wilde, Angela D Minic, Lee Niswander |
| Muscular Dystrophy Association | MDA-69338 | M Gartz Hanson |
| National Institute of Neurological Disorders and Stroke | F32-NS059267 | M Gartz Hanson |

The funders had no role in study design, data collection and interpretation, or the decision to submit the work for publication.

### Author contributions

MGH, Conception and design, Acquisition of data, Analysis and interpretation of data, Drafting or revising the article; JJW, RLM, Acquisition of data, Analysis and interpretation of data; ADM, Contributed significantly to the data design and acquisition, Acquisition of data, Analysis and interpretation of data, Contributed unpublished essential data or reagents; LN, Analysis and interpretation of data, Drafting or revising the article

### Ethics

Human subjects: Ethical approval was obtained through University of Colorado COMIRB approval for human subject research (protocol number 12-1504). All samples were provided through an exempt IRB which stated that we were disallowed patient information that could be used as an identifier. Animal experimentation: All experiments were conducted in accordance with the protocols described in the Guide for the Care and Use of Laboratory Animals (NIH. Revised 2011). All of the animals were handled according to approved institutional animal care and use committee (IACUC) protocol (B-69913(10)1D) of the University of Colorado Denver.

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
