## [Decision Letter]

Thank you for sending your work entitled “Rescue of Central Core Disease in mouse through potassium dependent mechanisms” for consideration at *eLife*. Your article has been evaluated by a Senior editor, a Reviewing editor, and 3 reviewers.

All three reviewers concur on the novelty and potential impact of your work. However they raise substantial concerns on the specificity of the phenomenon described and, even more importantly, on the electrophysiological mechanism leading to the pathology.

Finally the manuscript requires a significant revision both in terms of addressing specific issues (e.g. why a given muscle was chosen) and a major editing for a high number of “incorrect or misleading statements, irrelevant references and poorly described results”.

Reviewer # 1:

This is an interesting report on a RyR1 mutant mouse model in which various effects of the mutation have been demonstrated in vitro and in vivo. It is a well presented and detailed study with therapeutic implications. Specificity is touched on but not addressed by the study. It is important to know if the changes observed are specific to this mutant of RyR1 and if similar changes occur when other genes that cause a myopathy are mutated.

Additional points to address:

1) CCD is a well defined clinical and pathological entity caused by mutations in RyR1. What other genes, as implied in the Introduction, can cause it? Cores, as a pathological feature, can be associated with defects in several genes but these disorders are not central core disease.

2) Not all cases of CCD show central nuclei, although they can be a feature and the number variable. The central/internal nuclei in the H&E figures are difficult to see in the images that appear to have a lot of dirt and blemishes.

3) Why was soleus chosen for most of the studies but images of femoral muscles shown in Figure 1 and the rectus femoris of treated mice? Similarly in Figures 7 and 9 soleus is not shown.

4) What was the pathology in the diaphragm? Diaphragm is mentioned in the Results but not in the Methods.

5) In the Methods, immunolabelling is mentioned. No immunolabelling appears in the Results.

6) The cores shown in Figure 1 are not as clearly delineated as in typical CCD and both fibre types are affected. A typical feature of CCD is uniformity of fibre type and a slow fibre profile of most fibres. Fibre typing is mentioned in the Discussion but differences between fibre types are not addressed. In Figure 7 the increase in COX may be a reflection of fibre typing, which should be addressed.

7) Second paragraph in the Discussion section: a feature of cores in CCD is the absence of mitochondria, not the reduced activity. Activity elsewhere in the fibre is usually normal.

8) Z line streaming in Figure 1 is not apparent, and the arrow in 1K is not indicating the Z line streaming. In 1L the structure shown is not clearly a T–tubules. These are usually only apparent at triads so showing the rest of the triad would be more informative. In addition T–tubules do not usually show any internal content.

Reviewer # 2:

This work provides a novel hypothesis to explain the skeletal muscle function defects associated with central core disease and proposes corresponding therapeutic options. Using a new mouse model that presents hallmarks of CCD, the authors suggest that diseased muscle cells suffer from defective mitochondrial function and ATP production, decreased intracellular K^+^ concentration due to over–activity of K_ATP_ channels, depolarized resting membrane voltage. Authors propose that altered RyR1 function decreases ATP production which opens K_ATP_ channels, leading to a decreased resting membrane potential which reduces membrane excitability and contributes to loss of force. Increased K^+^ diet or K_ATP_ channel inhibition rescues histological defects, intracellular K^+^ level, mitochondrial function defects and muscle performance. Data from human muscle samples suggest that increased expression of several K^+^ channels may be used to biomark CCD.

This is an impressive study in terms of amount of collected data with a quite amazing diversity of techniques mastered by the authors. The rescuing effects of the K^+^ diet and K_ATP_ channel inhibition on the defects observed in the diseased muscles also appear quite spectacular. While the overall general trend of data tends to convincingly highlight that there is a problem of membrane excitability associated with the diseased muscle model and that there is therapeutic potential for treating this problem, there are two major issues that I find do considerably weaken the manuscript.

My first major concern relates to the model proposed by the authors to interpret their data. I first find it difficult to believe that a 30 % decrease in ATP (Figure 3), supposedly caused by reduced RyR1 Ca^2+^ release, would be sufficient to open K_ATP_ channels. But even assuming so, I find it even harder to understand how an increased K_ATP_ channel activity produces a chronic membrane depolarization whereas the opposite would be expected. Increased K^+^ conductance should favor a resting voltage where net K^+^ flux across the membrane is limited, so why K_ATP_ channel opening in this model produces a decreased intracellular K^+^ concentration and a membrane depolarization is very hard to understand. Similarly, how can increasing extracellular K^+^ lead to hyperpolarization is also counter–intuitive. The authors propose that diseased muscle cells are in the depolarized paradoxal P2 state, but it is completely unclear how this can be due to the increased K_ATP_ channel activity as one would rather expect a P2 state to result from an increased inward cationic conductance. This whole interpretation is very confusing and not convincing.

My second major concern is that the manuscript suffers from a number of incorrect or misleading statements, irrelevant references and poorly described results.

*Reviewer # 3:* The authors have identified a mutation in RyR1 that shows hallmarks of central core disease (e.g. muscle weakness). The finding is interesting, but the physiological/pathophysiological conclusions are unfortunately lacking scientific bearing.

Major concerns:

1) In Figure 2: *RyR1*^*AG*^ fails to release Ca^2+^ after depolarizing the membrane. As mentioned by authors, DHPR (Ca_v_1.1) is responsible for coupling to RyR1, which then releases Ca^2+^. The authors need to verify that the E–C coupling is functioning and that the complete lack of release is not due to impaired DHPR–RyR1 coupling. In Figures 1 and 3 it appears as 4–CMC can release some Ca^2+^ from SR via RyR1, hence the complete lack of Ca^2+^ release in Figure 1 seems not only be caused by impaired RyR1 gating. This needs to be further investigated by the authors. For example, muscle contraction is frequency dependent. The authors should add data showing the frequency–Ca^2+^ relationship.

2) Figure 2 shows that myotubes are not contracting; however, myotubes are immature cells that rarely contract in response to electrical stimulation. Instead the authors need to do these experiments on adult muscle fibers. Instead of expressing cells as “% contracting cells”, the data should be presented as factional shortening.

3) In Figure 2, the authors stated that membrane depolarization does not result in Ca^2+^ hence does not make any contractions possible. However in Figure 1 the mice appears to be able to use their muscles. In Figure 1 the authors measure force with grip strength and hanging score, unfortunately these methods are not specific to force production but also involve action potential quality and other parameters. The authors need to express force produced in a more specific way (e.g. ex vivo measurements) and also measure the cross–sectional area of the fibers to verify that the lack of force is not due to decreased muscle size.

4) In Figure 3, the authors have moved into isolated adult soleus muscles which is a slow–twitch fiber type. Please add pictures of the soleus muscles loaded with Rhod–2 or MitoSox.

5) In Figure 4, are diaphragm muscles used as well as soleus? Diaphragm is of mixed muscle fibers, do they also show the same mitochondrial parameters as soleus? Do fast–twitch muscle fibers also show signs of muscle weakness?

6) Rhod–2 measures mitochondrial Ca^2+^ and Figure 3 shows less Ca^2+^ uptake in the mutant muscle. The authors claim that this is due to mitochondrial defects, but the authors needs to verify that this is not reflecting less Ca^2+^ is released from RyR1?

7) To draw the conclusion that there is no Ca^2+^ leak based on no difference in MitoSox signal is not significantly correct since Ca^2+^ leak needs to be measured in an alternative way. Ca^2+^ leak should be assessed with Ca^2+^ imaging e.g. with Mag–Fluo or another low affinity dye. In addition, how is the MitoSox fluorescence normalized? A positive control is necessary. H202 can be used to substrate inhibit SOD which could give indications of mitochondrial ROS handling.

8) Figure 3 shows a TMRE fluorescence that assesses mitochondrial membrane potential. Please explain why Saponin was added to the fibers. In addition, in order to get an idea of the membrane potential the fibers should be treated with FCCP in the end of the experiment and then normalized to this value.

---

## [Author Response]

As suggested, we have performed new experiments and obtained additional data to support and extend our observations. These new data include additional functional analyses of the mutant RyR1 and ratiometric analysis of both Ca^2+^ and K^+^. Combined with our original findings, these new data indicate that the *Ryr1*^*AG*^ mutation inhibits the proper uptake of SR Ca^2+^ and release of SR Ca^2+^ from RyR1 channels and suggest that the *Ryr1*^*AG/+*^ channels are hyposensitive channels that decrease the myoplasmic calcium levels. We also show that *Ryr1*^*AG/+*^ muscles have a persistent K^+^ outward leak that can be overturned with the addition of external K^+^ or inhibition of K_ATP_ channels. We no longer suggest that this is a model for CCD because no corresponding mutation in humans has yet been reported in the literature and instead classify this model as a myopathy with core-like structures. We have cleaned up the data as the reviewers have suggested and corrected errors in the text and references.

*Reviewer # 1*:

*This is an interesting report on a RyR1 mutant mouse model in which various effects of the mutation have been demonstrated* in vitro *and* in vivo*. It is a well presented and detailed study with therapeutic implications. Specificity is touched on but not addressed by the study. It is important to know if the changes observed are specific to this mutant of RyR1 and if similar changes occur when other genes that cause a myopathy are mutated*.

We agree with the reviewer that it will be interesting to know whether other mouse models show similar changes due to potassium treatments. However, we feel this is beyond the scope of our current manuscript that provides a thorough mechanistic understanding of a novel allele of RyR1 and extends the mechanistic insights to therapeutic strategies and to samples from human patients with CCD myopathy. At the end of the Discussion we have raised this important point: “In the future, it will be of interest to determine whether the positive response to increased potassium levels is applicable to myopathies due to other RyR1 mutations and to other genetic causes.”

*Additional points to address*:

*1) CCD is a well defined clinical and pathological entity caused by mutations in RyR1. What other genes, as implied in the Introduction, can cause it? Cores, as a pathological feature, can be associated with defects in several genes but these disorders are not central core disease*.

In consideration of this comment, we have decided not to classify this mouse model as CCD but instead have used a broader term of a “myopathy with core-like structures”.

*2) Not all cases of CCD show central nuclei, although they can be a feature and the number variable. The central/internal nuclei in the H&E figures are difficult to see in the images that appear to have a lot of dirt and blemishes*.

We have quantified the number of central/internal nuclei in the text and Figure 6. These central nuclei are readily detectable in the mutant muscle whereas we detected none in the wildtype myofibers, and hence we consider central nuclei to be part of the myopathic phenotype. However, as noted by the reviewer, classification of CCD is difficult and therefore we have changed the description to a myopathy with core-like structures.

*3) Why was soleus chosen for most of the studies but images of femoral muscles shown in*
Figure 1
*and the rectus femoris of treated mice? Similarly in*
Figures 7 and 9
*soleus is not shown*.

We choose the vastus lateralis for all mouse histology shown in panels. This was an error in writing from a previous submission in which more panels were used. We have corrected this. As stated in the second paragraph of the Results section: “Because proximal muscles of limbs are typically taken for muscle biopsies in patients, we examined the vastus lateralis and adductor magnus. Additionally, we examined the soleus muscle as it is commonly used for in vitro studies of fatigue and Ryr1-associated myopathy in mouse models.” We also chose the soleus because it is an easily accessible muscle that is impacted in myopathies. In addition, the soleus is a flattened muscle, which is advantageous for the inside-out path clamp experiments and for confocal microscopy visualization for the PBFI and Fura-2 AM studies. Moreover, the soleus was the predominant muscle examined in the *RyR1*^*I4895T*^ and *RyR1*^*Y522S/WT*^ mouse mutants for ultrastructural studies (8; 80; 7). We chose to examine multiple muscles because myopathies are diseases of limb muscle weakness, not weakness of a particular muscle. Moreover, we examined multiple muscle types for gene expression by RT-PCR because the muscle chosen for patient muscle biopsies can be variable and the site of biopsy is unknown to us under the terms of our IRB protocol. Using multiple assays and different muscles with different fiber type compositions, our data are consistent and show muscle weakness, pathological changes, and gene expression changes. The consistency of our data between muscle types further strengthens our hypotheses and conclusions and should not be seen as a detriment to our manuscript.

*4) What was the pathology in the diaphragm? Diaphragm is mentioned in the Results but not in the Methods*.

We have removed the diaphragm data to avoid confusion with another muscle examined. Nonetheless, the diaphragm data are consistent with the phenotype described.

*5) In the Methods, immunolabelling is mentioned. No immunolabelling appears in the Results*.

We are sorry for this error; it has been corrected.

*6) The cores shown in*
Figure 1
*are not as clearly delineated as in typical CCD and both fibre types are affected. A typical feature of CCD is uniformity of fibre type and a slow fibre profile of most fibres. Fibre typing is mentioned in the Discussion but differences between fibre types are not addressed. In*
Figure 7
*the increase in COX may be a reflection of fibre typing, which should be addressed*.

There are numerous papers in the literature using NADH-TR staining that show cores are not always clearly delineated, such as the recent paper by Romero et al. in a patient with eccentric core disease caused by a mutation in *MYH7* (2014, JNNP) that shows some delineated cores and some non-delineated cores in the same sample. We quantified the cores in Type 1 muscle fibers at 12 months (Table 1) which were comparable to the NADH-TR staining delineations as Zvaritch (2009, PNAS). Cores were not present at 2 months of age so we are not reporting the fiber types within the muscle and instead focused on internalized nuclei (a hallmark for muscle damage, potassium and calcium levels and phenotypic weakness). Because of the difficulty of defining CCD, we no longer suggest that this is a model for CCD and instead classify the histological changes within the muscle as a myopathy with a core-like structure that can be delineated and non-delineated.

*7) Second paragraph in the Discussion section: a feature of cores in CCD is the absence of mitochondria, not the reduced activity. Activity elsewhere in the fibre is usually normal*.

Our data clearly show regions of mitochondrial dysfunction and regions of apparently normal mitochondria by histology, COX and NADH-TR staining, and mitotracker labeling. However, we cannot definitively say that mitochondria are absent in cores, and therefore we have chosen to use the term mitochondrial dysfunction.

*8) Z line streaming in*
Figure 1
*is not apparent, and the arrow in 1K is not indicating the Z line streaming. In 1L the structure shown is not clearly a T–tubules. These are usually only apparent at triads so showing the rest of the triad would be more informative. In addition T–tubules do not usually show any internal content*.

We have better placed the arrows in Figure 1 to identify the Z line streaming errors. Moreover, our ultrastructural analysis in the 2-month soleus muscle is similar to the study of I4895T and Y522S RyR1 mutant soleus (2 month) by [8] and [7]. Thus, the histopathological and ultrastructural disturbances that we observe in our mouse model are similar to I4895T and Y522S RyR1 mutations used to model human CCD.

*Reviewer # 2*:

*My first major concern relates to the model proposed by the authors to interpret their data. I first find it difficult to believe that a 30 % decrease in ATP (*Figure 3*), supposedly caused by reduced RyR1 Ca*^*2+*^
*release, would be sufficient to open K*_*ATP*_
*channels*.

First we would like to point out that ATP level usually decreases by only 20% during strenuous exercise in muscle fibers (63; 39; 40; 36). Thus, any additional decrease in ATP may alter other ATP dependent mechanisms. Second, 90% of ATP is used for muscle contractions by the myosin ATPase, Ca^2+^ ATPase and Na^+^/K^+^ ATPase (34; 18; 64; 5; 17). With such limited amounts for other muscle processes (K_ATP_ channels in particular), a 30% loss might be expected to have a significant effect on fiber force. Third, we also show that other regulators of K_ATP_ channel activity (AMPK, SUR1) are upregulated in the *Ryr1*^*AG/+*^ muscle, which would be expected to increase K_ATP_ channel activity. Fourth, we have functional data using pharmacological methods that shows an increase in intracellular K^+^ upon K_ATP_ channel inhibition. These data along with our new data showing a potassium leak (new Figure 4) and disrupted Ca^2+^ homeostasis through ratiometric imaging of myoplasmic Ca^2+^ in *Ryr1*^*AG/+*^ muscle (new Figure 2) provide strong mechanistic insights. We have now included comments related to all of these points within the Discussion, as well as new data and figures to substantiate these points.

*But even assuming so, I find it even harder to understand how an increased K*_*ATP*_
*channel activity produces a chronic membrane depolarization whereas the opposite would be expected. Increased K*^*+*^
*conductance should favor a resting voltage where net K*^*+*^
*flux across the membrane is limited, so why K*_*ATP*_
*channel opening in this model produces a decreased intracellular K*^*+*^
*concentration and a membrane depolarization is very hard to understand*.

*Similarly, how can increasing extracellular K*^*+*^
*lead to hyperpolarization is also counter–intuitive. The authors propose that diseased muscle cells are in the depolarized paradoxal P2 state, but it is completely unclear how this can be due to the increased K*_*ATP*_
*channel activity as one would rather expect a P2 state to result from an increased inward cationic conductance. This whole interpretation is very confusing and not convincing*.

In response to this question and those of reviewer 3, we have performed additional experiments that help to resolve this confusion but which have also led to the decision to remove the membrane potential sharp electrode recordings from the paper. Inhibition of K_ATP_ channel activity raises intracellular potassium levels in wildtype muscle, suggesting that K_ATP_ channels either efflux K^+^ or increase the permeability of K^+^ ions across the membrane of the muscle as a function of potassium homeostasis. In heterozygous mutant muscle we show increased expression and activity of K_ATP_ channels (Figure 5) and in Figure 4 we show a decrease in intracellular K^+^ over time, which can be reversed by inhibition of K_ATP_ channels. Upon performing additional sharp electrode recordings using a standard method, we found that the fresh soleus muscle underwent contractions during dissection, mounting and the introduction of the sharp electrode. We believe that the contractions, in combination with the potassium leak, slowed down the re-establishment of the proper membrane potential in *Ryr1*^*AG/+*^ muscle. When a mutant muscle was dissected, mounted and cultured for 1 hour at room temperature in 3 mM K ringers solution prior to sharp electrode recording, the muscle only showed a 5 mV polarized shift. While this data is potentially interesting, the interpretation of the data is difficult to explain using the techniques at our disposal, and thus we have removed the membrane potential sharp electrode recordings from the paper.

*My second major concern is that the manuscript suffers from a number of incorrect or misleading statements, irrelevant references and poorly described results*.

Thank you for your suggestions and corrections. We have made these corrections, as well as substantially revised the manuscript for clarity, in addition to the inclusion of new data to further substantiate the most important points of the paper.

*Reviewer # 3*:

*The authors have identified a mutation in RyR1 that shows hallmarks of central core disease (e.g. muscle weakness). The finding is interesting, but the physiological/pathophysiological conclusions are unfortunately lacking scientific bearing*.

*Major concerns*:

*1) In*
Figure 2*:* RyR1^AG^
*fails to release Ca*^*2+*^
*after depolarizing the membrane. As mentioned by authors, DHPR (Ca*_*v*_*1.1) is responsible for coupling to RyR1, which then releases Ca*^*2+*^*. The authors need to verify that the E–C coupling is functioning and that the complete lack of release is not due to impaired DHPR–RyR1 coupling. In*
Figures 1 and 3
*it appears as 4–CMC can release some Ca*^*2+*^
*from SR via RyR1, hence the complete lack of Ca*^*2+*^
*release in*
Figure 1
*seems not only be caused by impaired RyR1 gating. This needs to be further investigated by the authors. For example, muscle contraction is frequency dependent. The authors should add data showing the frequency–Ca*^*2+*^
*relationship*.

We have removed the original data in Figure 2 about homozygous mutant myotubes and replaced this with a new Figure 2 on ratiometric calcium imaging of heterozygous muscle, which show dysfunction of the RyR1 mutant channel.

*2)*
Figure 2
*shows that myotubes are not contracting; however, myotubes are immature cells that rarely contract in response to electrical stimulation. Instead the authors need to do these experiments on adult muscle fibers. Instead of expressing cells as “% contracting cells”, the data should be presented as factional shortening*.

We have removed the original Figure 2 (embryonic myotubes from homozygous mutants) and now show adult muscle from heterozygous animals (new Figure 2).

*3) In*
Figure 2
*the authors stated that membrane depolarization does not result in Ca*^*2+*^
*hence does not make any contractions possible. However in*
Figure 1
*the mice appears to be able to use their muscles. In*
Figure 1
*the authors measure force with grip strength and hanging score, unfortunately these methods are not specific to force production but also involve action potential quality and other parameters. The authors need to express force produced in a more specific way (e.g. ex vivo measurements) and also measure the cross–sectional area of the fibers to verify that the lack of force is not due to decreased muscle size*.

Original Figure 2 has been removed. As for examination of ex vivo force contractions, we show that the mutant RyR1 channel is hyposensitive (new Figure 2), suggesting a problem at the muscle, and we use the clinical examination for muscle weakness, which is an acceptable method for examining weakness.

*4) In*
Figure 3*, the authors have moved into isolated adult soleus muscles which is a slow–twitch fiber type. Please add pictures of the soleus muscles loaded with Rhod–2 or MitoSox*.

Added Rhod-2 pictures to Figure 3.

*5) In*
Figure 4*, are diaphragm muscles used as well as soleus? Diaphragm is of mixed muscle fibers, do they also show the same mitochondrial parameters as soleus? Do fast–twitch muscle fibers also show signs of muscle weakness?*

Diaphragm data has been removed to avoid confusion between limb and diaphragm muscle. We originally examined the diaphragm because it is a flattened muscle that was easy to visualize. Nonetheless, our current manuscript uses multiple assays and different muscles with different fiber type compositions to show consistent evidence for muscle weakness, pathological changes, and gene expression changes. The consistency of our data between muscle types further strengthens our hypotheses and conclusions.

*6) Rhod–2 measures mitochondrial Ca*^*2+*^
*and*
Figure 3
*shows less Ca*^*2+*^
*uptake in the mutant muscle. The authors claim that this is due to mitochondrial defects, but the authors needs to verify that this is not reflecting less Ca*^*2+*^
*is released from RyR1?*

We now show new data suggesting less calcium release from RyR1 (Figure 2). Thank you for the comment.

*7) To draw the conclusion that there is no Ca*^*2+*^
*leak based on no difference in MitoSox signal is not significantly correct since Ca*^*2+*^
*leak needs to be measured in an alternative way. Ca*^*2+*^
*leak should be assessed with Ca*^*2+*^
*imaging e.g. with Mag–Fluo or another low affinity dye. In addition, how is the MitoSox fluorescence normalized? A positive control is necessary. H202 can be used to substrate inhibit SOD which could give indications of mitochondrial ROS handling*.

Antimycin A was used as a positive control to compare intensity of fluorescence but was inadvertently left out of the original manuscript. It has been added to the revised Methods.

*8)*
Figure 3
*shows a TMRE fluorescence that assesses mitochondrial membrane potential. Please explain why Saponin was added to the fibers. In addition, in order to get an idea of the membrane potential the fibers should be treated with FCCP in the end of the experiment and then normalized to this value*.

As now stated in the Results section, saponin was added “to avoid the potentially confounding influence of variations in plasma membrane potential”, because even small variations in the plasma membrane potential caused by the potassium leak may change the interpretation of the data. FCCP was also performed in the cultures but was inadvertently omitted from the original submission. It has been added to the revised Methods section.